# Non-invasive in vivo sensing of bacterial implant infection using catalytically-optimised gold nanocluster-loaded liposomes for urinary readout

Kaili Chen[1], Adrian Najer [1,2]✉, Patrick Charchar [3], Catherine Saunders[1], Chalaisorn Thanapongpibul[1], Anna Klöckner[1,4,5], Mohamed Chami[6], David J. Peeler[1,5,7], Inês Silva[7], Luca Panariello[1,8], Kersti Karu [9], Colleen N. Loynachan [1], Leah C. Frenette[1], Michael Potter[1], John S. Tregoning [5], Ivan P. Parkin [9], Andrew M. Edwards [4,5], Thomas B. Clarke [4,5], Irene Yarovsky [3]✉ & Molly M. Stevens [1,7,8]✉

*Staphylococcus aureus* is a leading cause of nosocomial implant-associated infections, causing significant morbidity and mortality, underscoring the need for rapid, non-invasive, and cost-effective diagnostics. Here, we optimise the synthesis of renal-clearable gold nanoclusters (AuNCs) for enhanced catalytic activity with the aim of developing a sensitive colourimetric diagnostic for bacterial infection. All-atom molecular dynamics (MD) simulations confirm the stability of glutathione-coated AuNCs and surface access for peroxidase-like activity in complex physiological environments. We subsequently develop a biosensor by encapsulating these optimised AuNCs in bacterial toxin-responsive liposomes, which is extensively studied by various single-particle techniques. Upon exposure to *S. aureus* toxins, the liposomes rupture, releasing AuNCs that generate a colourimetric signal after kidney-mimetic filtration. The biosensor is further validated in vitro and in vivo using a hyaluronic acid (HA) hydrogel implant infection model. Urine samples collected from mice with bacteria-infected HA hydrogel implants turn blue upon substrate addition, confirming the suitability of the sensor for non-invasive detection of implant-associated infections. This platform has significant potential as a versatile, cost-effective diagnostic tool.

*Staphylococcus aureus*, including methicillin-resistant *S. aureus* (MRSA), are responsible for a raft of minor and serious infections in humans and is the leading cause of surgical implant-associated infection[1]. Surgical site infections (SSI) are a major complication associated with procedures involving indwelling medical devices[2,3]. The infection of implanted devices typically results in hard-to-treat biofilm formation and necessitates removal and replacement,

significantly increasing morbidity, mortality and cost[4,5]. A major challenge in combatting this problem is that many infections become established by the time they are detected. Therefore, developing new approaches to achieve earlier diagnosis of bacterial infection is key to addressing this problem. To identify infection at the site of an implant, it is necessary to collect samples from patients using swabs, aspirations, and even tissue biopsy or removal of the implant[6,7], which is

invasive, challenging and carries the potential risk of causing additional infections. Traditional bacterial culture is the gold standard detection method but requires a relatively long incubation time with limited specificity[7]. Advanced methods have been developed to complement bacterial culture for a comprehensive diagnosis with higher sensitivity and specificity, including polymerase chain reaction (PCR), enzyme-linked immunosorbent assay (ELISA), and mass spectrometry (MS)[8–10]. However, the sampling is invasive and can introduce the risk of contamination on the implants or the specimens, leading to false-positive results. Additionally, these methods require highly specialised equipment and extensive lab-based technical support, resulting in high costs and limiting their applications in resource-limited laboratories and medical institutions. Hence, there is a high demand for a non-invasive, in-situ approach for detecting implant-associated infection.

Leveraging the enzymatic or membrane-damaging activities of bacteria-derived virulence factors, including pore-forming toxins, enzymes (lipase activity), surface proteins (surfactant-like mechanisms) and adhesion proteins, is an emerging and promising alternative method to detect bacterial infection[11,12]. This provides a foundation for the design of various non-invasive diagnostic tools for the detection of *S. aureus*[13–15]. The toxin action has been harnessed for creating liposomal systems responsive to bacterial growth for detecting infections[16,17]. Liposomes with a cell membrane-like composition, especially formulated by utilising bacterial lipase-sensitive sphingomyelin lipids in the composition, are commonly used in specific drug delivery and sensing applications for bacterial infection[18–23].

Ultra-small (1–5 nm in size) gold nanoclusters (AuNCs) have recently become prominent in the fields of drug delivery[24,25], photothermal therapy[26], bioimaging[27,28] and biosensing[29,30], due to the advantage of their biocompatibility and rapid clearance through the urinary system compared to larger gold nanoparticles (>10 nm)[31–33]. Urinary readout after administration of various types of nanosensors is particularly attractive for diagnostic purposes due to its simplicity and non-invasiveness and it has previously been employed to detect implant rejection, bacterial infection, and cancer[32,34–37]. In our previous work, the catalytic activity of AuNCs was used in an in vivo sensing application for cancer, employing urinary readout[32]. In tumour-bearing mice, AuNCs covalently linked to a protein carrier via enzyme-responsive peptides were liberated and excreted into urine upon enzyme cleavage in vivo. The AuNCs catalyse the hydrogen peroxide ($H_2O_2$)-mediated oxidation of a chromogenic substrate resulting in a colour change visible to the naked eye. This approach facilitates the development of non-invasive, real-time monitoring systems for early diagnosis and personalised treatment strategies. Improvement in the catalytic activity of the underlying AuNCs could enhance sensitivity of the diagnostic to allow more accurate and earlier detection of pathological conditions. However, the AuNC size must remain below the glomerular filtration cut-off (ca. 5.5 nm)[38] to ensure rapid clearance of catalytically optimised AuNCs through the kidneys. Mechanistic understanding of stabilising ligand distribution and behaviour around the AuNCs is another key aspect required to move forward the development of diagnostic tests using renal-clearable, catalytically active AuNCs.

Here, we significantly improved the sensitivity of the AuNC sensing platform by optimising both the catalyst and carrier design, and expanded its utility to new disease areas. We designed an AuNC-based sensing system for detecting bacterial implant infection that assembles a large number of AuNCs (>100) per liposomal sensor unit and responds to pathogenic bacterial growth in the vicinity. First, we optimised the catalytic activity of AuNCs (renally clearable with a size of 1–2 nm) to improve our previously reported materials[32] by varying the synthesis conditions. The optimised AuNCs achieved enhanced signal amplification and they were characterised using a variety of experimental and computational techniques. This improvement in AuNC catalytic performance can enhance the sensitivity of the sensing

platform, allowing for detection even with minimal liberation of AuNCs, which is particularly relevant when aiming to detect disease early. Compared to our previous work[32], we improved the catalytic activity of AuNCs, changed the sensor design away from a covalent peptide-AuNC conjugate to a non-covalent liposomal system, expanded the application from cancer to bacterial infection, and provided extensive mechanistic understanding by leveraging various single-particle techniques and computational analysis. Here, the AuNCs were subsequently encapsulated in 100 nm liposomal nanocarriers, which could be disassembled by exposure to toxins secreted by pathogenic bacteria as characterised by single particle-sensitive methods. Hyaluronic acid (HA) hydrogels, clinically used as dermal fillers, were loaded with the optimised sensor and in this context served as a model for implant-associated bacterial infections. We demonstrated a successful response of our biosensor to *S. aureus* growth in contaminated HA hydrogel implants in vitro. A mouse model confirmed the usability of our sensor for the non-invasive detection of *S. aureus* implant infection in vivo with a simple urinary readout, serving as a proof of concept for Point-of-Care early detection methods. Our sensing platform takes advantage of the toxin activity to trigger the disassembly of liposomes in combination with the catalytic activity and renal clearance ability of inorganic AuNCs. This versatile biosensing platform, featuring a colourimetric readout in urine, achieved rapid and non-invasive detection of implant-related bacterial infection. The system highlights its significant potential for revolutionising healthcare practices, paving the way for impactful advancements in infection detection and broader healthcare improvements.

## Results
### Catalytic activity optimisation of AuNCs
Catalytic AuNCs were synthesised through reduction of gold (III) chloride trihydrate by glutathione (GSH), which were subsequently encapsulated within liposomes to create a sensing system (Fig. 1a). The action of bacterial toxins on the lipid membrane released AuNCs in situ. The catalytic activity of the AuNCs directly serves as an indicator of bacterial infection state. This detection leverages the use of sphingomyelin lipids in the liposome formulation, which can be hydrolysed by the toxins secreted by *S. aureus*. AuNCs capped with GSH were selected as probes due to their low cytotoxicity, high stability against harsh environments, high signal amplification with a high surface-to-volume ratio, stable catalytic activity in complex physiological environments, and small size that allows rapid and efficient renal clearance. *S. aureus*-contaminated or bacteria-free HA hydrogels with AuNC-loaded liposomes were intraperitoneally (i.p.) implanted into mice to serve as an in vivo model of implant infection detection (Fig. 1b). AuNCs were liberated from the liposomal sensor unit by exposure to toxins produced by nearby pathogenic bacteria (Fig. 1c). The AuNCs rapidly reached the kidneys and were filtered into the urine since they are smaller than the renal filtration threshold of 5.5 nm[32,38]. A colourimetric assay utilising 3,3′,5,5′-tetramethylbenzidine (TMB) oxidation through AuNC catalysis was performed in the collected urine, producing a blue colour change to verify the presence of AuNCs as an indicator of bacterial infection (Fig. 1d).

First, we optimised the synthesis of nanoclusters to enhance their catalytic property. We explored various conditions including difference in core metals (gold and platinum), core metal-to-thiol ratios, reducing agents and solvents (Supplementary Table 1). The peroxidase-like catalytic activity of AuNCs was assessed by measuring the absorbance at 652 nm, which resulted from TMB oxidation via $H_2O_2$ catalysed by AuNCs[39], represented throughout as kinetics, endpoint absorbance, or onset velocity. Through comparative analysis, clusters based on gold (Au) were found to exhibit superior catalytic activity compared to platinum (Pt) under various conditions. The nanoclusters synthesised in a weak reducing condition led to faster increase in absorbance over time, revealing enhanced catalytic activity,

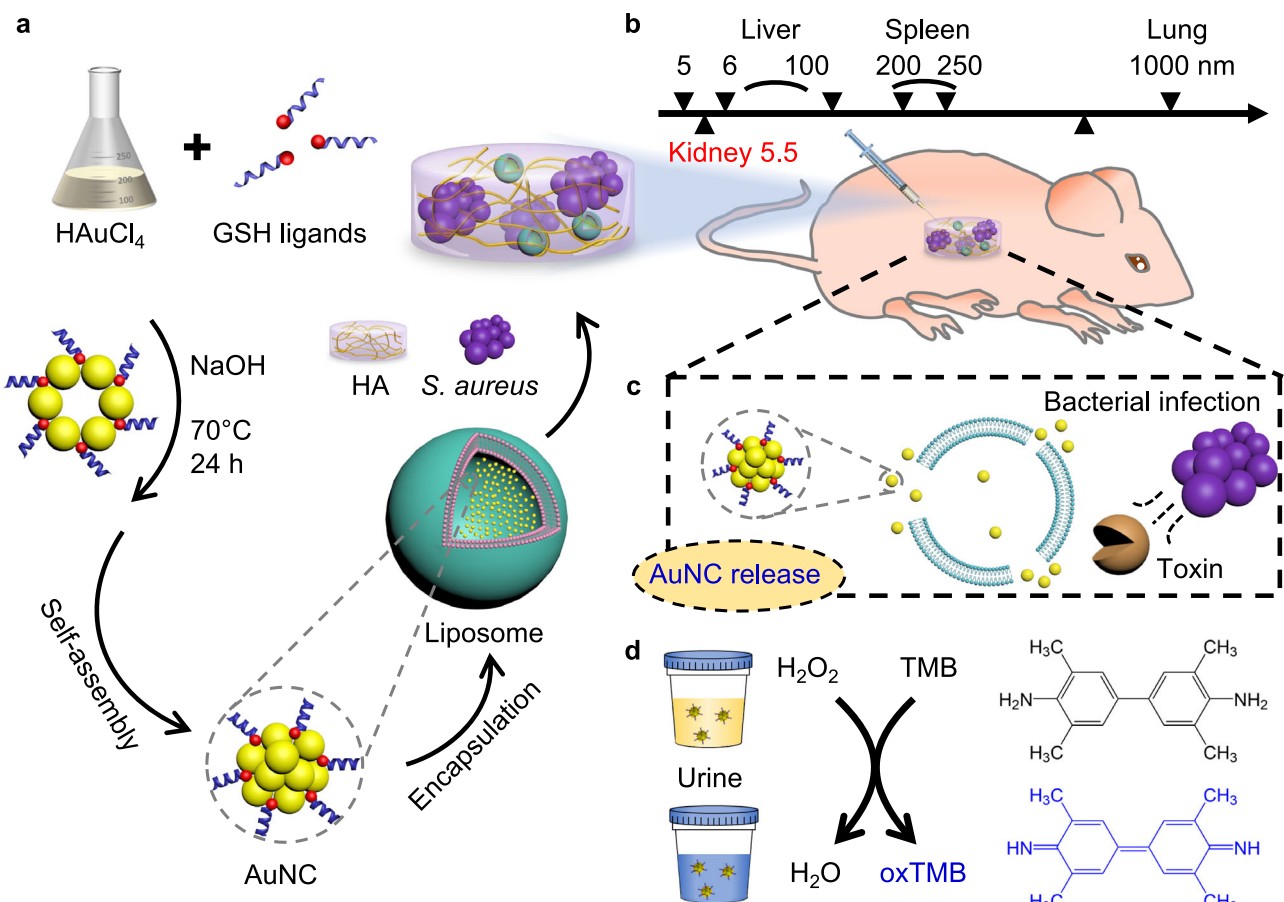

**Fig. 1 | Design of AuNC-loaded liposome biosensors and mechanism for detecting bacterial implant infection. a** AuNC synthesis. Au (III) was reduced to Au (0) to self-assemble as GSH-stabilised AuNCs, which were further encapsulated in liposomes to create the sensing system. HA hydrogels were used as model implants, embedded with AuNC-loaded liposomes with or without bacterial contamination. **b** Size threshold of particle clearance from different organs[32,41]. The ultra-small AuNCs were below the kidney filtration threshold (5.5 nm)[32,38] and were

able to be rapidly cleared into the urine. **c** Response mechanism of AuNC-loaded liposomes sensing system in presence of pathogenic bacteria growth concomitant with the secretion of lipid membrane-damaging toxins. **d** Colourimetric TMB oxidation assay in collected urine. The colour change of TMB substrate from colourless to blue demonstrated the presence of AuNCs which is the indicator of bacterial infection state of the implant.

whilst a change in solvent from aqueous to organic (ethanol) for synthesis diminished catalytic activity (Supplementary Fig. 1). The variation in synthesis temperature and metal-to-GSH ratio affected the size and catalytic activity of nanoclusters, with 70 °C and Au:GSH ratio of 1:1 and 1:1.5 representing optimal conditions for catalytic enhancement, whilst maintaining the size below the renal clearance cut-off of 5.5 nm (Supplementary Fig. 2). The effect of the ratio of metal-to-GSH on size is in agreement with previous literature, with nanocluster diameter decreasing with the increased ratio[40]. For these optimised 1:1 and 1:1.5 samples, we compared their sizes using transmission electron microscopy (TEM) images and analysed their catalytic functionality in different physiological environments such as in foetal bovine serum (FBS) and urine, both at 50 v/v % (Supplementary Fig. 3). Their small size was confirmed by TEM ( < 5.5 nm). Compared to measurements in PBS, >80% of catalytic activity was retained, even after 12-h incubation in physiologically relevant environments highlighting the exceptional stability of this AuNC composition.

We also optimised the synthesis pH for AuNC synthesis before encapsulating AuNCs into liposomes. Notably, AuNCs with metal-to-GSH ratio of 1:1.5 synthesised at pH 10 exhibited the highest activity, which was approximately 33.5-fold greater than the 1:1.5 AuNCs synthesised at pH 2 reported in our previous paper[32] (Fig. 2a–c). TEM images and the histogram derived from measurements indicate that the average size of the final optimised AuNCs synthesised in basic

conditions (pH 10) (1.7 ± 0.4 nm) falls below the renal clearable threshold (5.5 nm), thereby enabling their efficient filtration through the urinary system[32,38,41] (Fig. 2a, b). The limit of detection (LoD) when studying a dilution series of purified AuNCs in synthetic urine for both the optimised and previously studied AuNCs[32] was 0.028 pmol and 2.33 pmol, respectively (Supplementary Fig. 4). This indicates an 83-fold increase in sensitivity for the herein optimised clusters (Au-to-GSH 1:1.5). The broad impact of AuNCs with higher catalytic activity was demonstrated by formulating a thrombin sensor using peptide-linked AuNC-protein conjugates analogous to our previous study with non-optimised AuNCs[32]. The sensors developed with the optimised AuNCs synthesised at pH 10 showed 6x higher signal and 10x higher signal-to-noise (S/N) compared to our previous study, confirming the importance of catalytic optimisation on diagnostic sensitivity (Supplementary Fig. 4g–i). UV-Vis spectral analysis revealed a calibration curve for the concentrations of the optimised AuNCs, which was used for the quantification of different batches of AuNCs (Supplementary Fig. 5a, b). Thermogravimetric analysis (TGA) showcased that the percentage of weight loss (%) attributable to the organic ligands (GSH) in our previously studied (pH 2) and the optimised (pH 10) AuNCs was nearly identical. This observation suggests that the ligand density remained consistent when synthesised at an equivalent metal-to-GSH ratio and varying pH of synthesis conditions (Supplementary Fig. 5c). Matrix-assisted laser desorption/ionisation time-of-flight (MALDI-TOF)

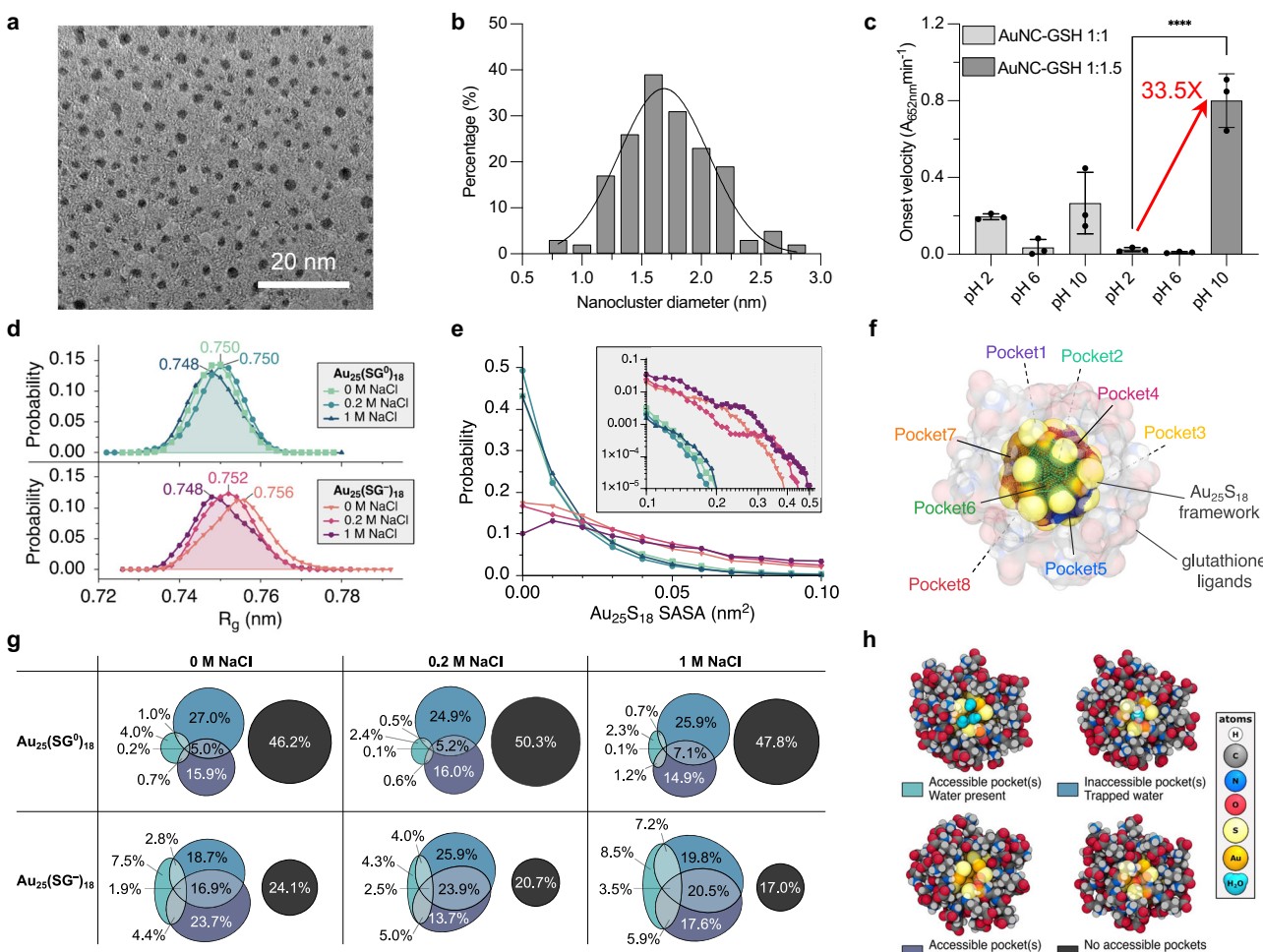

**Fig. 2 | Optimisation, characterisation, and simulation of AuNCs. a** TEM image of AuNCs (optimised synthesis at 1:1.5 Au-to-GSH ratio, pH 10) used in the liposome encapsulation. Scale bar 20 nm. **b** The size distribution histogram from TEM images of AuNCs in (**a**) (mean diameter = 1.7 ± 0.4 nm (s.d.), $n$ = 170 particles). **c** Catalytic activity of AuNCs after synthesis in different pH reaction conditions. The catalytic activity was calculated from the onset velocity ($A_{652nm}$ $min^{-1}$) when the TMB reaction kinetics had a linear trend (mean values ± standard error, $N$ = 3 independent biological replicates, $n$ = 3 technical replicates, One-Way ANOVA with Tukey post hoc test, $p < 0.0001$). **d**–**h** Mechanistic properties of glutathione-capped gold nanoclusters in varying salt and pH conditions, derived from all-atom molecular dynamics simulations. Probability distributions for the: (**d**) NC radius of gyration, $R_g$, and (**e**) the accessibility of the inner $Au_{25}S_{18}$ framework to small molecules, such as $H_2O$, $H_2O_2$, and TMB, calculated as solvent accessible surface area (SASA) with the corresponding molecular probe radius. A SASA of 0 $nm^2$ signifies none of the

inorganic NC core surface area is accessible. **f** Transparent rendering showing the location of the eight triangular active sites (i.e., pockets) on the internal gold–sulphur nanocluster surface. A graphic animation illustrating the dynamic opening and closing of an accessible active site can be found in Supplementary Movie 1. **g** Euler plots showing the probability to find one or more active sites on $Au_{25}(SG)_{18}$ accessible (binary open/closed, i.e., SASA > 0 $nm^2$) and with water in close proximity to the exposed metal atoms. Overlapping regions indicate that within a single $Au_{25}(SG)_{18}$ snapshot, multiple binding pockets have different configurations. Colours for each circle/oval match the atomic renderings shown in (**h**) to illustrate the four possible scenarios an individual catalytic pocket may be in: open with water present (lightest blue), closed with trapped water (medium blue), open without water present (dark blue), or closed without trapped water (black). Atoms are shown in a space-filling representation and coloured according to the inset key. Only solvent proximal to the relevant binding pocket is shown for clarity.

mass spectrometry analysis, which is often performed on such AuNCs[42,43], revealed that both our previously developed (pH 2) and the optimised (pH 10) AuNCs exhibited a similar molecular weight, ranging from 2000 to 12,000 Da, along with an identical Au-to-GSH ratio. This similarity suggests that the pH variation during synthesis may be associated with other surface properties, such as the ligand-to-Au bond or the Au oxidation state rather than a change in the ligand composition and/or size of the clusters (Supplementary Fig. 5d, e). To evaluate the catalytic activity of AuNCs in environments mimicking the usual assay conditions (acidic pH) and the in vivo situation (neutral pH), the catalytic activity of AuNCs was tested through TMB oxidation assays, varying pH levels and concentrations of sodium chloride (NaCl) (Supplementary Fig. 5f, g). Both cluster types showed the same decrease in catalytic activity at neutral vs acidic assay pH, whilst only the pH 2 not the pH 10 AuNCs revealed slightly elevated catalytic

activity at higher salt concentration at low pH, although the signal still remained much lower than for pH 10 synthesised AuNCs. Overall, these results suggest that AuNCs, particularly those with a specific metal-to-GSH ratio, alkaline pH synthesis, and acidic assay conditions, exhibit significantly improved catalytic performance compared to previous findings[32], which will allow the creation of more sensitive diagnostic tests for a variety of settings.

## Molecular dynamics insights into ligand mobility and active site accessibility of AuNCs

Using all-atom molecular dynamics (MD) simulations, we investigated how the mobility and arrangement of GSH ligands in different solution conditions affect the accessibility of active sites on $Au_{25}(SG)_{18}$ (Fig. 2d–h), a prototypical model for the synthesised ultrasmall AuNCs. Atomic level monitoring and understanding of the coverage/exposure

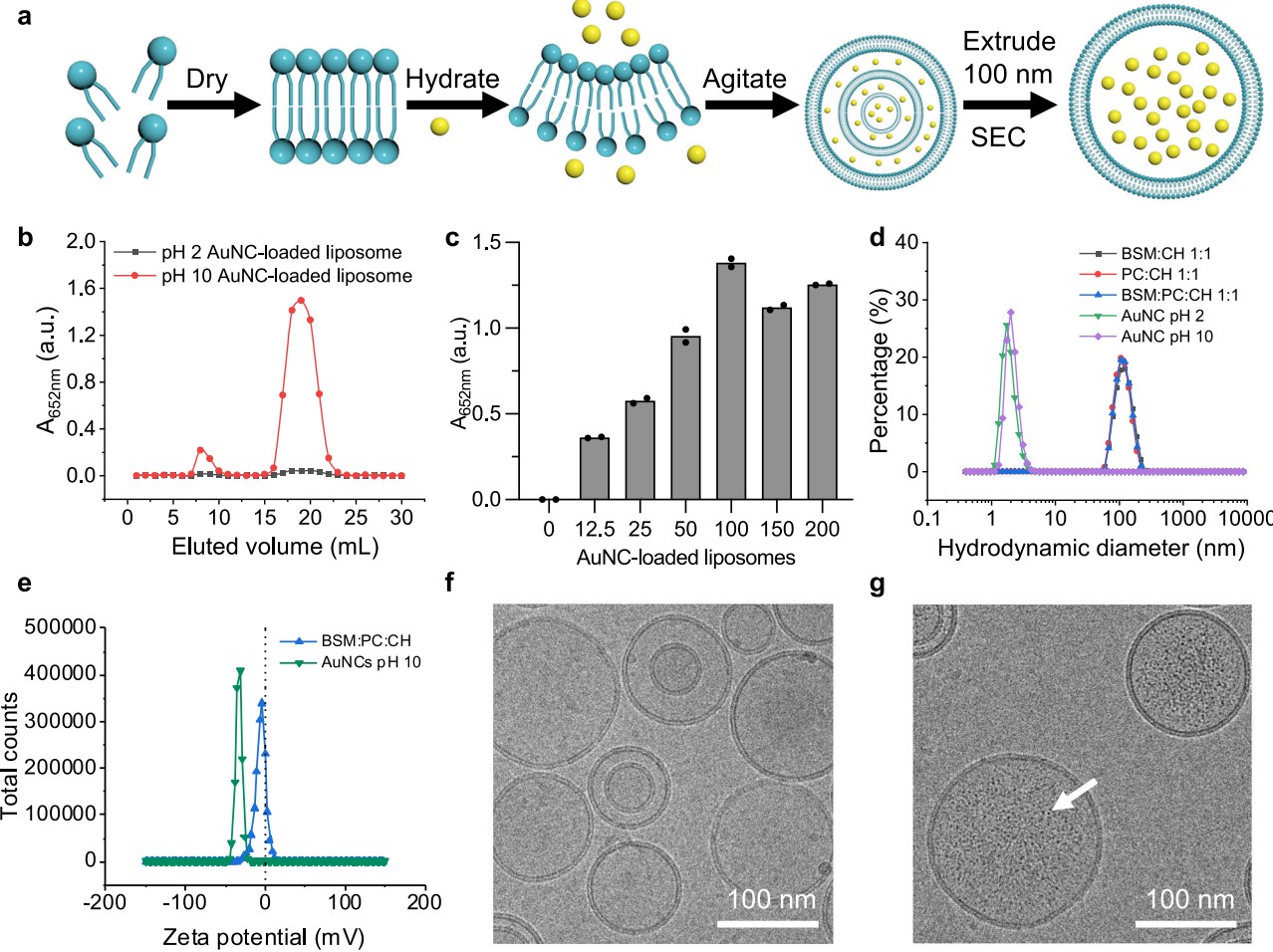

**Fig. 3 | The assembly and characterisation of AuNC-loaded liposomal sensor.**
**a** Schematic of the encapsulation of AuNCs in liposomes. The lipid components of liposomes include brain sphingomyelin (BSM), 1-palmitoyl-2-oleoyl-glycero-3-phosphocholine (16:0–18:1 PC), cholesterol (CH), 1,2-distearoyl-sn-glycero-3-phosphoethanolamine-N-[methoxy(polyethylene glycol)-2000] (DSPE-PEG2k), and 1,2-dioleoyl-3-trimethylammonium-propane (DOTAP) in various ratios. **b** TMB oxidation kinetics measurement on each eluted fraction from size exclusion chromatography (SEC) for separation of unloaded AuNCs from liposomes ($n = 1$ measurement). The larger liposomes were eluted first, followed by smaller unloaded AuNCs. Liposomes loaded with AuNCs pH 10 exhibited significantly higher catalytic activity in both fractions than those loaded with AuNCs pH 2. **c** Catalytic activity of all concentrated liposome fractions after SEC loading with different concentrations of AuNCs. Numbers refer to the times of concentration compared to AuNCs pH 10 1X (20 μM). Liposomes loaded with 100X of AuNCs pH 10 (2 mM) performed best, with the optimal catalytic activity according to the endpoint at 60 s ($n = 2$ technical repeats). **d** DLS measurement results of AuNCs (number distribution) and AuNC-loaded liposomes (intensity distribution) of different formulations using BSM, PC, CH and DSPE-PEG2k (average of $N = 1$ sample, $n = 3$ measurements). **e** Zeta potential distribution measurements of AuNCs pH 10 100X and the liposomes loaded with AuNCs 100X (curves are mean of $n = 3$ technical measurements). **f, g** Cryo-TEM image of empty liposomes (a total of 32 images were taken with similar results) (**f**) and AuNC-loaded liposomes (a total of 28 images were taken with similar results) (**g**) (BSM:PC:CH 1:1:1 with 1 mol % DSPE-PEG2k). Scale bar 100 nm. The white arrow indicates one loaded AuNC. Additional images and comparison to empty liposomes is shown in Supplementary Fig. 14.

of Au cluster surface sites is highly valuable due to its direct relevance to catalytic activity. We explored the exposure of Au cluster surface active sites under three NaCl salt concentrations (0 M, 0.2 M, and 1 M) and two GSH ligand protonation microstates, $Au_{25}(SG^0)_{18}$ and $Au_{25}(SG^-)_{18}$, to gain insights into the behaviour of the AuNCs in acidic (pH ~2.5) and neutral (pH ~7.5) environments, respectively. This information is extrapolated to the in vitro TMB assay (0 or 0.2 M salt, acidic pH) and in vivo physiological (0.2 M salt, neutral pH) conditions, where the AuNCs' polyprotic dissociation equilibria result in protonation microstates with a variable number of COOH/COO⁻ moieties. AuNCs with a combination of protonated and deprotonated ligands have been experimentally reported for aqueous[44] and gas-phase[45,46] GSH-capped AuNCs, as well as, computationally predicted for other aqueous peptide-coated $Au_{25}$ systems[47].

Here, our nanocluster models were constructed with non-equilibrium GSH ligand configurations (Supplementary Fig. 6) that

were then relaxed in ambient conditions with respect to the initial arrangements and equilibration was verified by monitoring the ligand atomic root-mean-square deviations (RMSD) and radii of gyration ($R_g$) (Supplementary Fig. 7–8). The observed RMSD plateaus and relatively small $R_g$ fluctuations suggested that the systems were exploring properties within the equilibrium phase space of the ambient conditions after 200 ns. The $R_g$ analysis also highlighted that the AuNC global shape is well preserved over the final 300 nanoseconds of MD for each trajectory (Supplementary Fig. 7–8), and that neutral pH conditions cause the ligand layer to slightly expand (see right-skewed $R_g$ for $Au_{25}(SG^-)_{18}$ in Fig. 2d). This expansion leads to a higher solvent accessibility of the internal $Au_{25}S_{18}$ framework for deprotonated $Au_{25}(SG^-)_{18}$ compared to protonated $Au_{25}(SG^0)_{18}$ (Fig. 2e). For $Au_{25}(SG^-)_{18}$, we find that 83–90% of NCs have an internal $Au_{25}S_{18}$ framework with a solvent accessible surface area (SASA) greater than 0 nm², of which approximately 8–18% of the structures are significantly

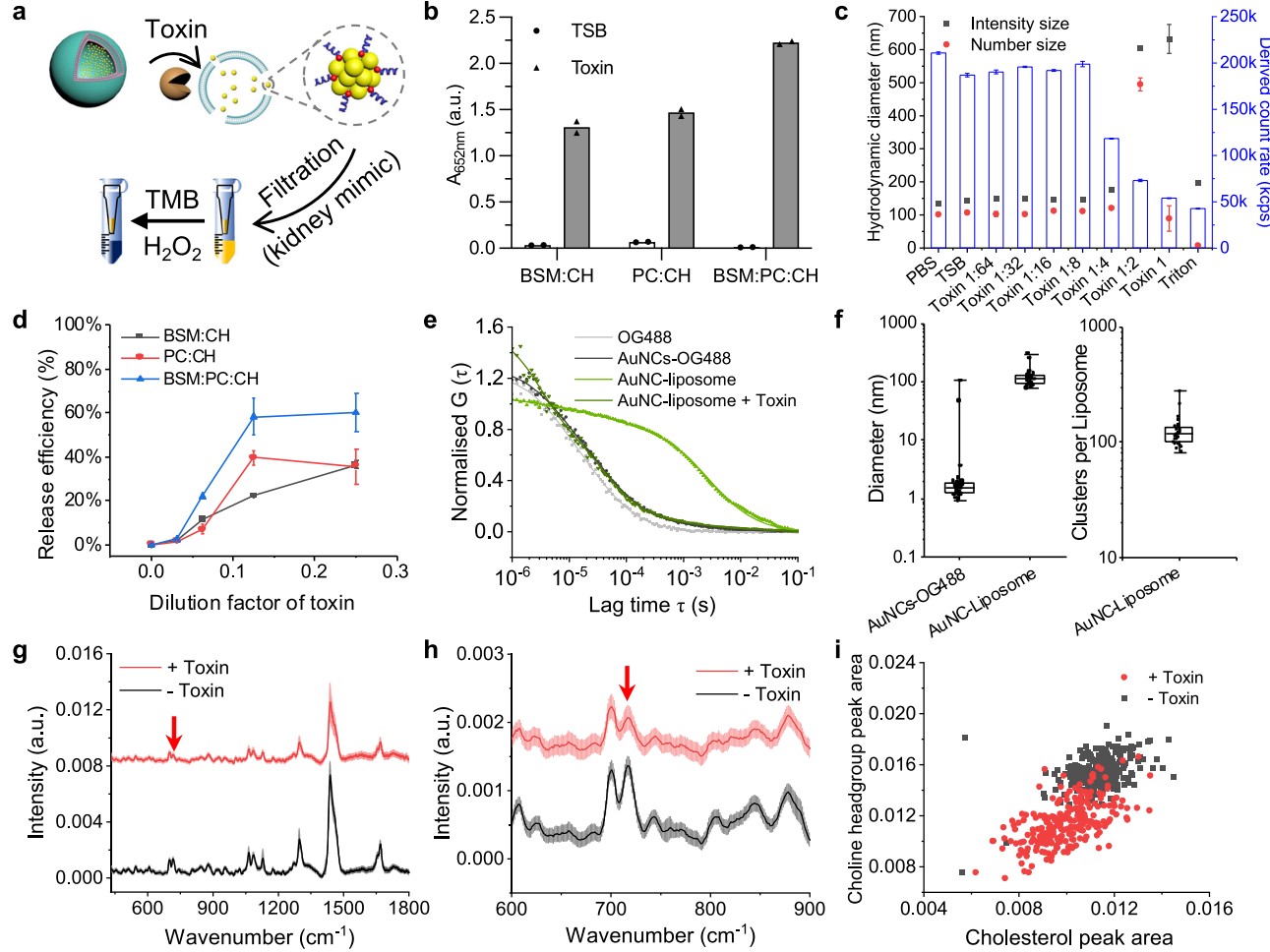

**Fig. 4 | In vitro bacterial toxin response of AuNC-loaded liposomes composed of different lipid formulations. a** Schematic of toxin incubation with AuNC-loaded liposome and separation of released AuNCs for TMB assay. **b** TMB catalytic activity assay on the filtrates of liposomes with different formulations for sensitivity comparison (1 mol % DSPE-PEG2k was used in all formulations as a stabiliser) with and without toxin incubation (mean values, $n = 2$ technical repeats). **c** DLS measurements of BPC (BSM:PC:CH) liposomes incubated with a dilution series of toxin, including hydrodynamic diameter (intensity and number distribution) and the derived count rates as an approximate guide of concentration (mean values ± standard error, $n = 3$ measurements). **d** The release efficiency determined via the filtrates after the incubation of the different liposomal sensors with various dilutions of *S. aureus* SH1000 overnight culture supernatant containing secreted toxins and centrifugal filtration to separate intact sensors from released AuNCs (mean values ± standard error, $N = 2$ biological replicates, $n = 1$ technical measurement). **e** Normalised average autocorrelation curves from FCS measurements of free dye

(OG488), dye-labelled AuNCs (AuNCs-OG488), liposomes loaded with AuNCs-OG488, and the toxin-treated liposomes (average curve of $n = 25$ individual measurements per sample). **f** Hydrodynamic diameters of AuNCs-OG488 and AuNC-OG488-loaded liposomes and the number of AuNCs per liposomes calculated from FCS molecular brightness analysis of data from (**e**). Box-plots: The centre line, the median; the box limits, the upper and lower quartiles; the whiskers, minimum and maximum values. **g, h** Mean single particle Raman spectra of AuNC-liposome before (black line) and after (red line) toxin treatment measured with single particle automated Raman trapping analysis (SPARTA®). Spectra were normalised to the area under the curve. The red arrow indicates choline headgroup peak at 717 cm⁻¹ ($n > 234$, the shading refers to S.D.). **g** refers to a full spectrum and (**h**) is a spectrum from 600 cm⁻¹ to 900 cm⁻¹. **i** Scatter plots of peak signal intensity from (**h**) of both samples at 700 cm⁻¹ (cholesterol peak area) and 717 cm⁻¹ (choline group peak area). These indicate the hydrolysis of sphingomyelin into ceramide and phosphorylcholine.

accessible (i.e., with a SASA greater than 0.1 nm²). In contrast, the Au₂₅(SG⁰)₁₈ simulations showed only 51–57% with a non-zero Au₂₅S₁₈ SASA (a decrease of 30%) and a negligible 0.4% of structures had a notably accessible core (>0.1 nm²). In the neutral pH setting, a higher number of open structures correlate with an increased salt concentration due to improved intermolecular electrostatic screening between the anionic glutathione ligands of Au₂₅(SG⁻)₁₈. These findings indicate that the deprotonation of GSH, especially within a high ionic strength environment, provides small molecules (e.g. H₂O, H₂O₂, and TMB) better access to the inorganic core of the NC.

Considering the mobility of the GSH ligands under the various pH and salt conditions, we assessed the impact of ligand movement on the entropic penalty for small molecules to access catalytic sites and/or the release of trapped water. It is well established that dynamics of GSH

ligands and water solvation shells are vital for an accurate description of glutathione-protected AuNCs[48,49]. Here, we observed through atomic root-mean-square-fluctuations (RMSF) that most ligand rearrangement is brought about by flexibility in the GSH peptides' C- and N-terminal regions as opposed to movement in the gold−sulphur interface (Supplementary Fig. 9). While we noted that sulphur atoms infrequently "flip" over the [Au−S−Au−S−Au] "staples" that protect the NC core[50], the overall ligand structure and active site accessibility was largely unaltered by these events.

On the icosahedral core of the Au₂₅(SR)₁₈ nanoclusters, there are eight equivalent Au₃ pockets (Fig. 2f) that have been previously identified as active sites for catalysis[51]. Using SASA, we monitored the dynamic opening and closing of the catalytic pockets (Supplementary Fig. 10). Irrespective of solution conditions (120 independent

simulations representing 3 μs of equilibrated data per system), at least one out of eight active sites[51] had a non-zero SASA and remained sterically accessible to small molecules such as water or hydrogen peroxide for 25–63% of the simulation time (Fig. 2g, h). We further refined our analysis by also monitoring the MD trajectories for the presence of water molecules in the vicinity of the catalytic sites (<0.5 nm to gold icosahedral core, Fig. 2g, h), where $H_2O$ (radius ~0.14 nm) acts as a proxy for $H_2O_2$ (radius ~0.18 nm) or TMB (length along the short semi-axis ~0.22 nm). The water population analysis revealed that the abovementioned configurations with accessible binding pockets (SASA > 0 $nm^2$) had water near the catalytic metal core for 4–25% of the simulation time (Fig. 2g–h). Interestingly, even when all active sites were covered by ligands and inaccessible to small molecules (SASA = 0 $nm^2$), an appreciable amount of trapped water (30–50%) remained on the cluster surface (Fig. 2g, h). In several cases, multiple catalytic pockets (out of eight) were open and/or had proximal water in a single $Au_{25}(SG)_{18}$ configuration. It is important to note that these statistics are based on instantaneous configurations recorded every 5 ps, while a water molecule may have moved into or out of a binding pocket in the time between the data points, or the ligands may have opened or closed the pocket. We therefore consider all observed scenarios where a binding pocket on $Au_{25}(SG)_{18}$ has a non-zero SASA and/or nearby water molecules to be a statistically significant opportunity for an active site to participate in peroxidase-like catalytic activity. These results emphasise that although pH and salt influence active site accessibility (Fig. 2g, h), the AuNC ligand layer undergoes molecular rearrangement irrespective of solution conditions to enable a small molecule access to the catalytic metal sites on the nano-to-microsecond timescale. In summary, the MD simulations highlight that while the GSH ligands protect the clusters to ensure stability in solution, they also continuously allow small molecule access to the active site(s), providing a plausible explanation for the consistent peroxidase activity of AuNCs in complex physiological environments.

## Preparation and characterisation of the liposomal sensor encapsulating AuNCs

To create the sensing unit, liposomes were self-assembled in the presence of aqueous solutions of AuNCs. The ratios of lipid components were varied and included brain sphingomyelin (BSM), 1-palmitoyl-2-oleoyl-glycero-3-phosphocholine (16:0–18:1 PC), cholesterol (CH), 1,2-distearoyl-sn-glycero-3-phosphoethanolamine-N-[meth-oxy(polyethylene glycol)-2000] (DSPE-PEG2k), and 1,2-dioleoyl-3-trimethylammonium-propane (DOTAP) (Fig. 3a). BSM was specifically chosen as a crucial component in liposomes due to the secretion of beta-toxin by *S. aureus*, which is a sphingomyelinase C[52,53]. This toxin turns sphingomyelin into ceramide, which induces negative membrane curvature, eventually causing liposome rupture[14,54]. To validate the effect of enhanced catalytic activity of AuNCs on the performance of this sensing platform, both optimised (1:1.5, pH 10) and previously developed AuNCs (1:1.5, pH 2 [32]) were encapsulated in liposomes and purified by size exclusion chromatography (SEC). To estimate the maximal signal that can be achieved after full release of AuNCs from the liposomes, Triton X−100 was added before running the TMB assay on the SEC fractions.

The liposome sensing system based on the optimised AuNCs exhibited a significantly higher catalytic activity after the lysis of the relevant liposome fractions (Fig. 3b; Supplementary Fig.11, 12). We further investigated the optimal concentration of AuNCs for loading by using inductively coupled plasma mass spectrometry (ICP-MS) to precisely measure the gold contents in liposomes after SEC purification. This helped to quantitatively determine the encapsulation efficiency of AuNCs within the liposomes. We discovered the liposome sensing platform loaded with an initial concentration of 2 mM AuNCs exhibited the best catalytic activity and with an encapsulation efficiency of 2.5% (Fig. 3c; Supplementary Fig. 13; Supplementary Table 2).

This was expected for passive loading of hydrophilic components (GSH-AuNCs), which ended up in the liposome core, generally reaching similar encapsulation efficiencies of a few percent limited only by the internal volume available. Dynamic light scattering (DLS) measurements of liposomes prepared with different formulations demonstrated consistency, with a hydrodynamic size of approximately 120 nm after extrusion using a 100 nm-pore membrane (Fig. 3d). Liposomes loaded with AuNCs revealed neutral to slightly negative zeta potentials as expected when incorporating 1 mol % DSPE-PEG2k (Fig. 3e; Supplementary Fig. 13b). The size and membrane structure of empty liposome controls and AuNC-loaded liposomes was confirmed by cryo-TEM (Fig. 3f, g; Supplementary Fig. 14).

## AuNC-loaded liposomes respond to toxins secreted by *S. aureus* in vitro

To assess the sensitivity of different liposome formulations to bacterial toxins, the liposomes were incubated with freshly prepared bacterial supernatant from *S. aureus* SH1000 strain overnight cultures containing the secreted toxins. *S. aureus* SH1000 is a laboratory strain, which is commonly used in research, and has the ability to secrete a variety of toxins. The incubation solution was subsequently filtered (100 kDa centrifugal filter to mimic kidney filtration) to collect the released AuNCs, and the filtrate was measured through the oxidation of TMB (Fig. 4a). This indicated the presence of released AuNCs and the sensitivity of the liposome sensing system. Among the tested formulations, liposomes composed of a 1:1:1 mass ratio of BSM, PC, CH and incorporating 1 mol % DSPE-PEG2k as stabiliser exhibited the highest release of AuNCs (Fig. 4b). The 1:1:1 lipid ratio mimics the base composition ratio found in red blood cell (RBC) membranes[55,56], which we selected since RBCs are a common tool for assessing toxin action, responding efficiently to toxins[57]. Importantly, *S. aureus* produces several toxins capable of lysing RBCs, making this lipid ratio an effective strategy for mimicry. Also, other ratios of these lipid components were used for comparison, demonstrating 1:1:1 was optimal in terms of its responsiveness to toxins (Supplementary Table 3, Supplementary Figs. 15, 16).

We hypothesised that additional cationic lipids in the formulation could improve the loading efficiency of anionic gold nanoclusters. A further attempt to optimise the liposome composition was conducted by incorporating cationic DOTAP lipid (positively charged) and performing comprehensive comparative characterisation and release quantification. This included assessment of zeta potential, nanoparticle tracking analysis (NTA) to compare concentrations, limit of detection (LoD), and release efficiency. To evaluate the release efficiency, Triton X-100 was used to fully lyse the liposomes, and all formulations were measured accordingly[58]. Considering both the surface charge of the liposomes and the toxin-triggered release signal amplification, we discovered that the inclusion of DOTAP did not contribute to a better performance of sensing. Liposomes composed of the above-described 1:1:1 mixture and incorporating 1 mol % DSPE-PEG2k exhibited high stability in bacteria medium (tryptic soy broth, TSB), high release efficiency of AuNCs in response to *S. aureus* toxin mixture, and the highest final signal after performing the catalytic reaction (Supplementary Figs. 16–18, Supplementary Table 4). The release efficiency reached 74% with the highest absorbance intensity on the released filtrate analysis when compared to other formulations. Hence, this formulation, termed BPC, was taken forward for subsequent experiments.

DLS measurements of BPC liposomes after incubating with different dilutions of toxin-containing bacterial culture supernatants revealed the impact on size and relative concentration (derived count rate as a rough estimate) of liposomes, which dropped when increasing the amount of toxin added (Fig. 4c). We used fluorescence correlation spectroscopy (FCS) to investigate the toxin-mediated release from the liposomes. FCS allows the characterisation of the diffusion

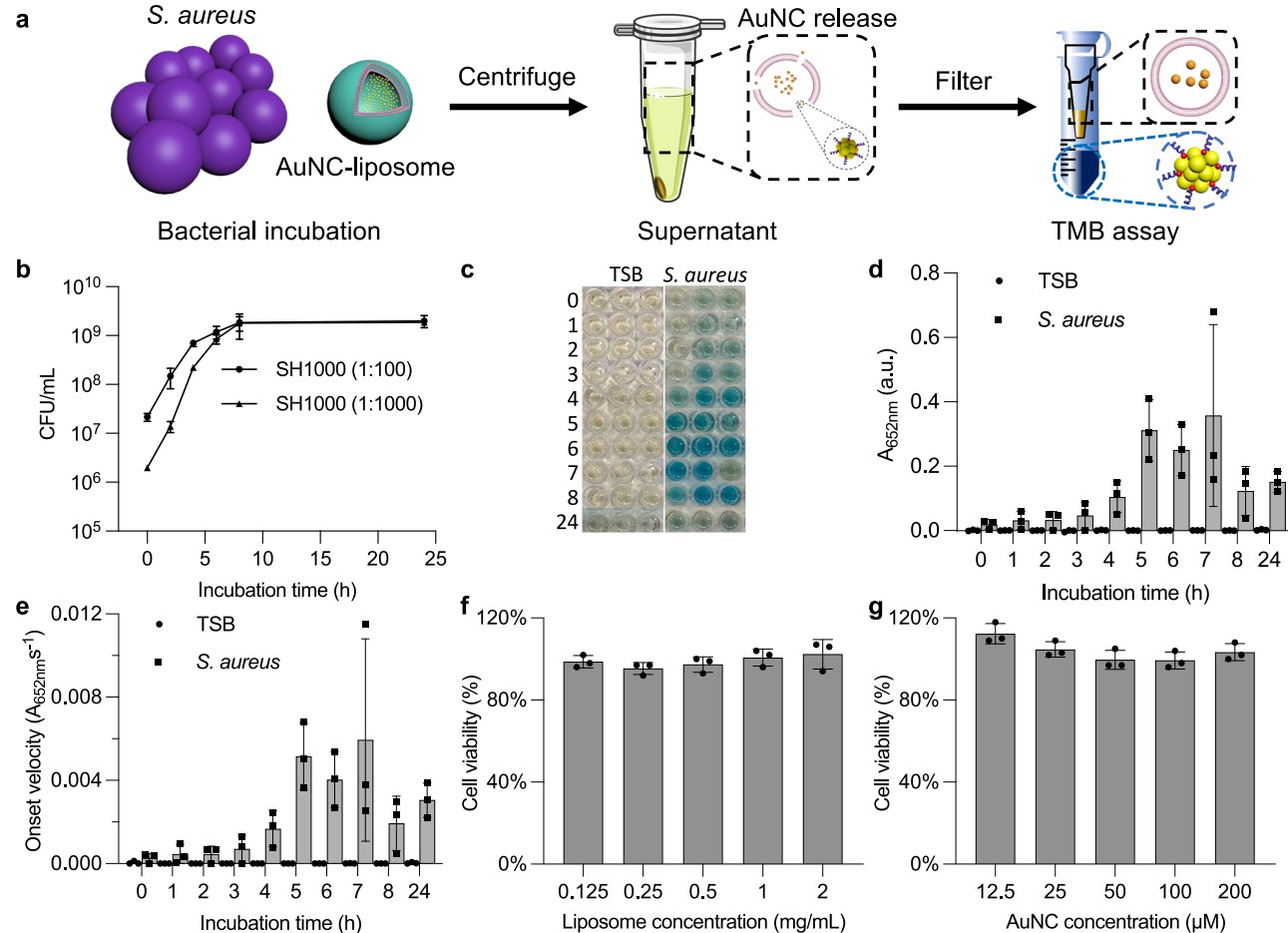

**Fig. 5 | In vitro incubation of bacteria with AuNC-loaded liposomes and characterisation of bacterial growth. a** Schematic of in vitro incubation of AuNC-loaded liposomes with bacteria and subsequent separation of released AuNCs from intact liposomes. **b** Bacterial growth curve of *S. aureus* SH1000 by CFU counting method (mean values ± standard error, $N = 3$ biological replicates, $n = 1$ technical repeat). **c**−**e** TMB oxidation assay on the filtrates after incubation of AuNC-loaded liposome with TSB ± *S. aureus* from 0 to 24 h. **c** Photographs of final TMB assay results. **d, e** Absorbance (at 652 nm at 140 s endpoint) and onset velocity for assay in **c**: TSB (black bars with circular dots) and TSB + bacteria (grey bars with square dots) (mean values ± standard error, $N = 3$ biological replicates, $n = 1$ technical replicate, additional repeats with detailed reaction curve in Supplementary Fig. 19a−e). **f, g** In vitro MTS cell viability assay after incubation of AuNC-loaded liposomes (**f**) and AuNC (**g**) with RAW 264.7 cells (mean values ± standard error, $N = 3$ biological replicates, $n = 3$ technical replicates).

properties (hence the hydrodynamic size) of a fluorescent sample and the amount of fluorescent cargo per liposome can be calculated using the molecular brightness information (counts per particle, cpp, in kHz). After incubation with toxin, the diffusivity of the liposomes completely shifted to that of freely diffusing OG488-labelled AuNCs and analysis indicated 99% release of AuNCs-OG488 upon 2-h toxin incubation, after correction for bright and slow diffusing species[59] (Fig. 4e). AuNC release as measured by FCS is higher than when measured via catalytic activity assessment (Supplementary Fig. 16), which can be attributed to the fact that FCS was performed without purification after toxin-incubation. The centrifugal filters used for purification prior to catalytic activity measurements have likely adsorbed a fraction of the released AuNCs. The hydrodynamic diameters of both the AuNCs-OG488 and AuNC-OG488-loaded liposomes calculated from the FCS autocorrelation analysis (Fig. 4f) were consistent with the sizes obtained from DLS. Brightness analysis (cpp) also revealed that in average >100 AuNCs were loaded per liposome, in agreement with the cryo-TEM images (Fig. 3g; Supplementary Fig.14).

To investigate the lipid transformations in liposomes exposed to toxin at a single-particle level, we employed the single particle automated Raman trapping analysis (SPARTA®) technique[60]. SPARTA® was recently reported as a label-free method to provide high-throughput

information about the precise chemistry compositions among a population of the nanocarriers and their cargo[61] and to measure time-resolved enzymatic conversion of lipid headgroups[62]. Here, SPARTA® measurements were performed on toxin-treated and untreated liposomal sensors. The mean Raman spectra across all liposomes exhibited similarities between the samples, with characteristic lipid peaks at 1298 cm$^{-1}$ and 1438 cm$^{-1}$ (CH$_2$ deformation), and 1670 cm$^{-1}$ (overlapping cholesterol, ceramide backbone, C = C). However, the signal intensity of toxin-treated liposomes was slightly lower than of the non-treated ones, likely due to less integrity after exposure to toxin (Fig. 4g, h). Notably, there was a reduction relative to the rest of the spectrum in intensity of the 717 cm$^{-1}$ peak after incubation with toxin, which can be attributed to the C-N symmetric stretching of the choline group[63]. This data confirms that the sphingomyelin lipids were hydrolysed into ceramide and soluble phosphorylcholine by sphingomyelinase-like toxin secreted from *S. aureus*[64]. The production of ceramide is known to cause membrane destabilisation and vesicle collapse resulting in cargo release[65,66], as observed here with AuNCs. After cleavage, the phosphorylcholine was likely released into solution, so it was no longer spatially associated with the liposomes, leading to the loss of the choline headgroup signal in the SPARTA® data after toxin incubation. This observation is consistent with previous

literature on the transformation of sphingomyelin to ceramide by sphingomyelinase[67]. Additionally, the peak signal at 700 cm$^{-1}$ is associated with cholesterol[63], which is a component present in the liposome formulation. The consistent presence of the stable signal at 700 cm$^{-1}$ was attributed to the cholesterol lipids in the liposome formulation, contributing to enhanced stability[68]. Scatter plots comparing the peak signal at 700 cm$^{-1}$ against the choline headgroup peak at 717 cm$^{-1}$ can be used to show the single particle-level variation of choline signal with cholesterol signal for both samples (Fig. 4i). Untreated liposomes formed a homogenous population with cholesterol and choline peak areas centred around 0.011 and 0.016, respectively. After toxin treatment, choline signal in the whole population notably decreased relative to the cholesterol signal which demonstrates relatively homogenous cleavage of the choline headgroup. Overall, the optimised liposome composition, BPC, was demonstrated to be responsive to toxin secreted from *S. aureus* in vitro and achieve an almost complete release of AuNCs, whilst the sensor remained intact when incubated without the toxins.

### AuNC-loaded liposomes respond to *S. aureus* in vitro

To assess the responsiveness of AuNC-loaded liposomes to bacteria in their vicinity – a more clinically relevant scenario – we conducted incubation experiments with *S. aureus* cultures instead of toxin supernatant as above. This approach aimed to mimic the sensing process during bacterial infection. Following incubation of the liposomal sensor with *S. aureus* cultures or with TSB only, the supernatant was obtained by centrifugation after different time points. The supernatant of each incubation was then filtered through a centrifugal filter (100 kDa) to collect the filtrate containing AuNCs released from the liposomes. The collected filtrates were subjected to TMB assays, where a colour change to blue indicated the release of AuNCs from the liposomes in the presence of bacterial growth (Fig. 5a).

To ensure consistent *S. aureus* SH1000 growth behaviour, an overnight bacterial culture was diluted at ratios of 1:100 and 1:1000, followed by subsequent serial dilutions. The diluted bacteria were streaked on tryptic soy agar (TSA) plates and allowed to grow overnight to obtain colony-forming units (CFU) for determining the bacterial concentration. The 1:100 diluted bacteria exhibited rapid growth in the initial 5 h and reached a maximum concentration of $10^9$ CFU/mL (Fig. 5b). The 1:100 dilution was taken forward for testing with the liposome sensor. The filtrates obtained from liposomes incubated with bacteria or TSB were mixed with the TMB substrate, and the absorbance at 652 nm was monitored to assess the kinetics of the reaction. The peak absorbance endpoint in the TMB assay indicated the catalytic activity of the released AuNCs, which directly correlates with the extent of their release. After a 3-h incubation, a blue colour change was observed in wells where liposomes were incubated with bacteria. This indicated the onset of liposome lysis by the bacterial toxins, which correlated with the endpoint measurements of the TMB assay. The catalytic activity reached its highest value after 5 h incubation with bacteria, as indicated from their absorbance and the onset velocity (Supplementary 19c, d). In contrast, the filtrates from liposomes incubated with TSB culture medium showed no colour change even after 48-h incubation, indicating the high stability of liposomes in media in the absence of bacteria (Fig. 5c–e; Supplementary Fig. 19d–f). The reaction curve kinetics also demonstrated a similar trend, with the fastest reaction observed at 5 h of incubation. The onset velocity of the reaction was consistent with the endpoint measurements, and reproducibility of the assay was confirmed (Fig. 5d, e; Supplementary Fig. 19a–e). The reason for reaching a plateau after 5 h was attributed the blockage of Amicon filter units caused by bacterial growth and the accumulation of liposome debris following disassembly. To further verify the stability of the liposomes, DLS characterisation was performed in different media. The results showed that the size of the liposomes remained around 120 nm without disassembly or aggregation after incubation in various media at body temperature over at least 48 h (PBS, PBS:TSB 1:1, and 10 v/v% human serum) (Supplementary Fig. 19g–i). To estimate the minimum amount of sensor that needs to be disassembled to achieve a detectable signal, the calibration curve for AuNC-loaded liposomes (BSM:PC:CH) was conducted and calculated to be 19 μg/mL of lipid in the presence of 86 pmol of AuNC (Supplementary Fig. 20).

An important consideration for the use of any biosensor is their biocompatibility. To assess the cytocompatibility of AuNC-loaded liposomes, RAW 264.7 cells were exposed to serial dilutions of the liposomal sensor (Fig. 5f) or AuNCs alone (Fig. 5g), and cell viability was evaluated using MTS assays. The results indicated that the sensor and the AuNCs exhibited excellent biocompatibility and the cells remained viable. This is in agreement with previous literature, liposomes of this composition are known for their biocompatibility[69] and GSH-AuNCs have also previously been tested and shown to be biocompatible in vitro and in vivo[32,70]. In conclusion, we synthesised a biocompatible sensor that responds to *S. aureus* growth in vitro.

### HA hydrogels as an implant model for the liposomal bacteria sensor

Various hyaluronic acid (HA) hydrogels have been approved for human use as injectable dermal fillers, with the advantage of high content of water, tissue-like properties and ease of implantability[71,72]. Here, we used a commercial HA hydrogel (Belotero Intense) as an implant model, which allows for simple physical loading of our liposomal sensor. These commercial HA hydrogels offer enhanced stability due to cross-linking technologies, which extend their duration and provide resistance to enzymatic degradation, ensuring a longer-lasting effect in tissue augmentation applications[73,74]. The commercial product we used (Belotero Intense) is widely used and has a reported longevity of up to 12 months after injection[75]. Infection of hydrogel dermal fillers through implantation of inadvertently contaminated implants is also a concern, similar to other implants[76]. To test the suitability of this implant system, free AuNCs or AuNC-liposomes were mixed with HA hydrogels and injected into PBS or TSB culture medium, incubated over time, followed by supernatant collection and TMB assay. These tests were performed to investigate the timescales of retention, diffusion, and release of free AuNCs and AuNC-liposomes out of the HA hydrogels. AuNCs readily diffuse out of the HA hydrogel within the first hour, demonstrating that liberated AuNCs from our sensor will be able to pass through the gel efficiently. In contrast, AuNC-liposomes were mostly retained within the hydrogel for up to 6 h of incubation at 37 °C and the centrifugal filter test confirmed that liposomes largely remained intact and the AuNCs remained encapsulated (Supplementary Fig. 21, 22). Overall, the HA hydrogel system was deemed suitable for our subsequent proof-of-concept tests in a mouse model of acute implant infection, as the liposomes largely remained within the hydrogel for the required timescale and the mixture remained readily injectable. Incorporating a covalent coupling strategy to anchor the sensor to the implant could extend the duration of liposome retention within/on the implant, but this will require extensive optimisation and we envisage this as future work.

Using the experimental setup described above, we then incorporated bacterial contamination into the system. AuNC-liposomes mixed with HA hydrogel were either bacteria-free or intentionally contaminated with *S. aureus*. The mixture was again injected into TSB for incubation, supernatant collection, and centrifugal filtration. The filtrates were collected for TMB assay to detect AuNCs released from the liposomal sensor (Fig. 6a). The sensor remained intact when incubated in bacteria-free HA hydrogels, whilst AuNCs were released from the sensor and detected by TMB assay for the bacteria-contaminated gels, as early as 1 h after the start of the incubation (Fig. 6b–d; repeat in Supplementary Fig. 22). Before in vivo administration, we confirmed efficient bacteria growth also in the presence of

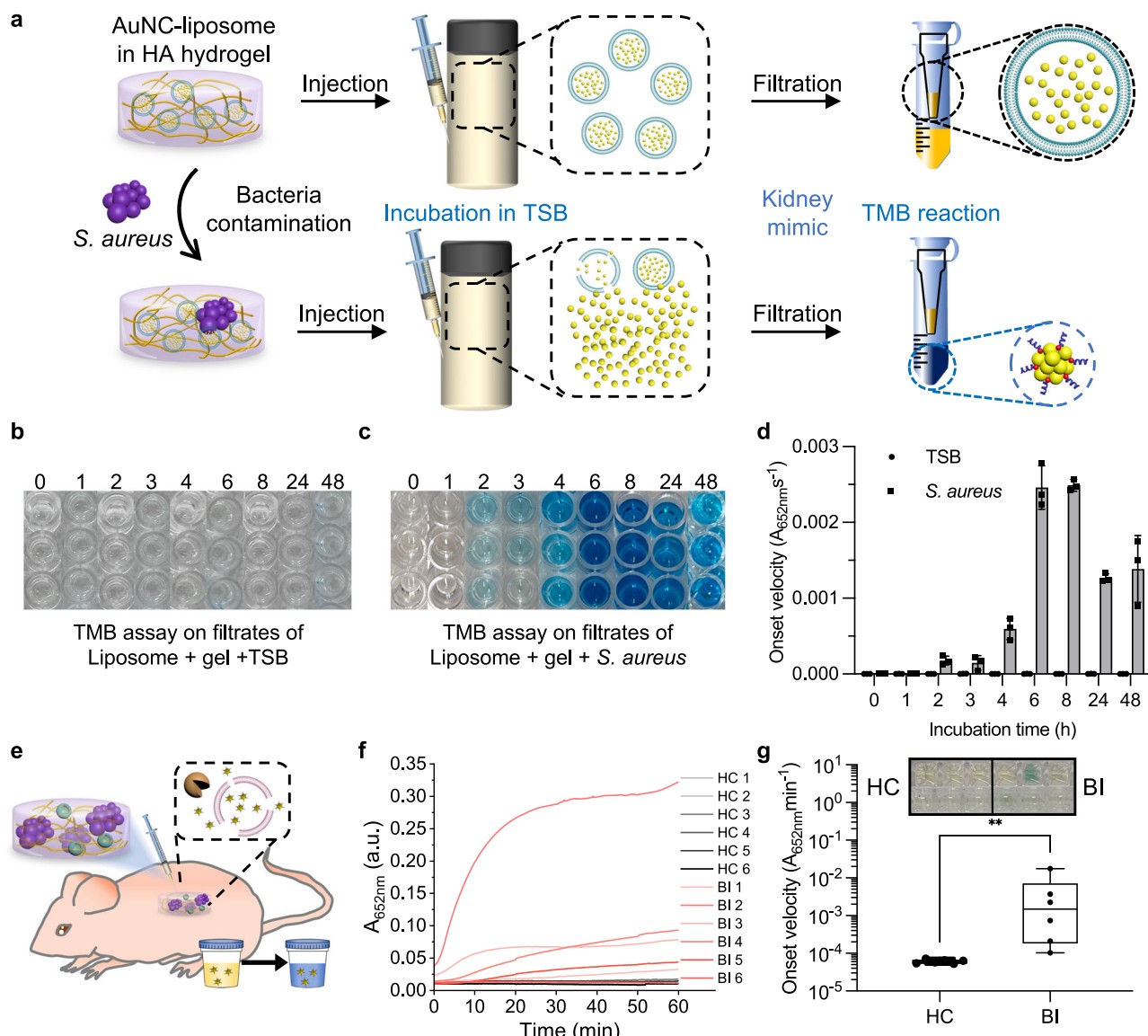

**Fig. 6 | AuNC-liposome sensing system enabling a colourimetric readout of bacterial infection state. a** Schematic of the HA hydrogel implant infection model and detection of in vitro released AuNCs. **b, c** Photographs of TMB colourimetric readout results for filtrates as shown in (**d**). **d** TMB assay on filtrates after incubation of the liposomal sensor mixed with HA gel ± bacteria, showing onset velocity. The incubation time varied from 0 to 48 h (mean values ± standard error, $N = 3$ biological replicates, $n = 1$ technical replicate, the corresponding absorbance measurements are shown in Supplementary Fig. 22b and the cleavage efficiency calculated from onset velocity and absorbance are shown in Supplementary Fig. 22c, d). **e** Schematic of mouse model of acute implant infection using i.p. administration of

HA-hydrogels loaded with AuNC-liposomes with or without *S. aureus* contamination. **f** TMB reaction kinetics measurements on the urine collected from mice 4 h after administration of sensor-loaded HA hydrogels with (bacterial infection, BI) or without (healthy control, HC) bacterial contamination with *S. aureus* SH1000 ($n = 6$ mice per group). **g** Onset velocity calculated from TMB catalytic activity assay in (**f**) ($n = 6$ mice per group, two-tailed Mann–Whitney test, $p = 0.0022$). Box-plots: The centre line, the median; the box limits, the upper and lower quartiles; the whiskers, minimum and maximum values ($N = 6$ biological replicates, $n = 1$ technical replicate).

HA hydrogels. Bacteria incubated in TSB with or without HA hydrogel had the same growth curve (Supplementary Fig. 23a–c). CFU counts and the absorbance at 600 nm (OD$_{600}$) were identical over time (Supplementary Fig. 23b, c). When considering the bacteria numbers, we can estimate that about $10^8$ CFU/mL produced enough toxin within 1–2 h to cause sufficient sensor rupture and AuNC release to be available for TMB signal generation. To evaluate the specificity of the liposomal sensor, we incubated sensors with supernatants from cultures of *S. aureus* and the non-pathogenic, non-toxin producing bacterium *Lactococcus lactis*. The TMB assay results indicated that the liposomal sensor specifically responded to the *S. aureus* toxin, as evidenced by the blue colour change observed in the filtrate containing released AuNCs. In contrast, no blue colour was visible in the filtrate

from the incubation with *L. lactis*, demonstrating that the liposomal sensor selectively responded to *S. aureus* (Supplementary Fig. 23d–h). This is in agreement with a previous study that revealed specificity of liposomal rupture in presence of pathogenic *S. aureus* vs non-pathogenic *L. lactis*[22]. We have further selected thrombosis as an unrelated disease to investigate diagnostic specificity. After incubation of the liposomal sensor with thrombin, no colour change was detected when performing the TMB assay in the filtrate (Supplementary Fig. 23d–h), confirming the inability of the sensor to respond to this enzyme. If future research reveals the need to improve sensor specificity further, we envision that a dual-response system with an enzyme-cleavable PEG shell could be explored. This strategy has previously been successful in improving specificity in the anti-cancer drug

delivery field[77]. Hence, the HA hydrogel implant model was found to be a suitable system to verify our bacteria sensor in vivo, as the implant is prone to bacterial infection and growth.

### AuNC-liposome biosensor responds to bacteria growth in vivo and provides simple urinary readout

GSH-AuNCs at a size of around 2 nm have been extensively studied with regard to biocompatibility, biodistribution, clearance through the kidneys into the urine, and catalytic activity retention in urine[32]. This previous work demonstrated rapid elimination of similar nanocluster compositions into the urine after i.v. administration using the TMB assay to verify their presence. To confirm renal clearance of our bare AuNCs also after i.p. administration, urine samples were collected from mice injected i.p. with AuNCs. The catalytic activity of cleared AuNCs was monitored by TMB oxidation reaction. PBS control injections gave no signal, whilst the urine samples after AuNCs i.p. injection and TMB reaction turned blue indicating successful clearance of AuNCs into the urine (Supplementary Fig. 24). The urine sample from 3 h post injection exhibited higher catalytic activity than that from 1 h, after normalisation for different dilution factors (less urine was available for 3-h sample, hence it was tested at a higher dilution than the 1-h sample). A calibration curve of onset velocity of AuNCs in urine was measured for the clearance efficiency of the AuNC clearance at 1 h and 3 h as 6.3% and 2.5% (not corrected for differences in urine volume). This data provides further insights into the renal clearance dynamics of bare AuNCs in mice via different routes of administration[78,79]. In previous work, the cleared AuNCs in urine were measured with up to 73% clearance efficiency compared to the injected dose after 1 h post injection by intravenous (i.v.) administration[32]. This suggests that AuNCs could be cleared through kidneys after i.p. administration but the clearance after i.p. administration was much slower than i.v. administration, as expected.

After confirming the in vitro response of our biosensor to bacterial infection in HA hydrogels and the successful clearance of AuNC through the kidneys in vivo, we employed an acute infection model in mice. *S. aureus* SH1000 strain has previously been used for the study of surgical bacterial implant infection[80–82]. We administered the implant with biosensor (AuNC-liposome-loaded HA hydrogels) to the model, either with or without bacterial contamination (*S. aureus* SH1000 strain), via i.p. route using two groups of six mice each: the healthy control group (HC) and the bacterial infection (BI) group. Urine collection was performed 4 h after the injection (Fig. 6e). The TMB catalytic activity kinetics assay was measured using 20 μL of the collected urine. The raw absorbance curves for the TMB reaction increased over time in the bacterial infection (BI) group (pink lines), whereas the healthy control (HC) group did not increase (grey, Fig. 6f). This confirmed the in vitro data, showing that only in the presence of bacterial contamination significant amounts of AuNCs were released from our sensor and cleared into the urine. Comparing the calculated onset velocities showed significantly higher signals from BI versus HC group (Fig. 6g). Additionally, we administered AuNC-loaded liposomes with HA gel to three healthy mice, along with two PBS injection controls, followed by TMB assays on urine samples collected over a 48-h period. The absence of blue colour change in both groups demonstrates that the sensing platform remained stable in vivo during that timeframe (Supplementary Fig. 25).

Further optimisation of AuNC catalytic activity, loading amount of AuNCs per liposome, and liposome composition are future means to further optimise the detection sensitivity. Future studies on the effect of variations in urine volume (dependent on liquid uptake) and time window of urine collection on signal intensity is required to define the necessary minimal sensor concentration needed at the implant site to achieve the envisioned binary yes/no readout of bacterial infection status. These results demonstrate that our biosensor, combining catalytic AuNCs and toxin-responsive liposomes, can realise rapid detection of implant infection with a colourimetric readout using a non-invasive method.

## Discussion

In this study, we have developed a biosensor platform as a rapid and sensitive colourimetric method for detecting bacterial infections within a short 4-h timeframe. Our research focused on the synthesis and optimisation of ultra-small AuNCs with significantly enhanced catalytic activity (33.5-fold higher than AuNCs in previous work[32], with 83-fold lower LoD), whilst accessibility of reactant molecules to the Au core was confirmed by MD simulations. The composition of the liposomes that encapsulated these AuNCs was varied to develop a biosensing platform with high sensitivity and responsiveness. We demonstrated that the biosensors could respond to toxins secreted by *S. aureus* in vitro. This particularly leverages the advantage that the secreted beta-toxin can hydrolyse the sphingomyelin lipid in the liposome formulation, thereby releasing encapsulated AuNCs, as verified by single-particle measurements. Finally, we validated the biosensor platform in solution, culture, and in hydrogels and successfully showed detection of released AuNCs in the urine from mice with a model of implant-associated bacterial infection. Overall, our work emphasised the significance of enhancing the catalytic activity of AuNCs due to the low encapsulation efficiency within liposomes and tailoring the liposome compositions to engineer a sensitive biosensor platform for the detection of bacterial infection in vivo, in which the catalytic activity of AuNCs can be used as a readout for disease monitoring. This development holds great promise for early detection of implant infection allowing for timely treatment to limit the chance of developing severe complications and other scenarios of bacterial infection. We envision the application of the current system for detecting acute implant infection. The clinical scenario could follow the same procedure as demonstrated in our animal experiments, involving mixing the sensor with the hydrogel before injection and collecting a urine sample from the patient a few hours after the procedure. If the urine turns blue upon addition of the TMB substrate solution, an antibiotic could be prescribed. Alternatively, the liposomal sensor could be injected locally in any area of suspected bacterial infection and urine could be collected at a defined later time point for sensing via the catalytic reaction. Further research and refinement of this biosensor platform could lead to its practical clinical application with particular relevance for at-home or resource limited settings due to its simple and sensitive colourimetric readout.

## Methods

All experimental data was plotted using Origin 10.0.0.154 and Prism 10, with schematics created using Autodesk 3ds Max and Power Point. Computational data was plotted using gnuplot (version 5) and molecular visualisations were rendered with Visual Molecular Dynamics (VMD, version 1.9.3)[83].

### Materials

Gold chloride trihydrate (HAuCl$_4$), chloroplatinic acid hexahydrate (H$_2$PtCl$_6$), L-glutathione reduced (GSH), sodium borohydride (NaBH$_4$), sodium hydroxide (NaOH), Ethanol (EtOH), NeutrAvidin Protein (NAv), 3,3′,5,5′-tetramethylbenzidine (1-Step™ Ultra TMB-ELISA Substrate Solution), 30% (w:w) hydrogen peroxide solution (H$_2$O$_2$), Triton X-100, Sepharose CL-2B, Gold Standard for ICP, TraceCERT(R), 1000 mg/L Au in hydrochloric acid, M-17 broth and thrombin from human plasma were purchased from Sigma-Aldrich. Brain sphingomyelin (BSM), 1-palmitoyl-2-oleoyl-glycero-3-phosphocholine (16:0–18:1 PC), cholesterol (CH), 1,2-dioleoyl-3-trimethylammonium-propane (DOTAP) and 1,2-distearoyl-sn-glycero-3-phosphoethanolamine-N-[methoxy(polyethylene glycol)−2000] (DSPE-PEG2K) were purchased from Avanti Polar Lipids. Methanol (MeOH) and chloroform (CHCl$_3$) were used as purchased. Dulbecco's phosphate buffered saline (DPBS),

DMEM without phenol red (1X, high glucose, GlutaMAX, Gibco, 31966-021) and Oregon Green 488 Carboxylic Acid, Succinimidyl Ester, 6-isomer (OG488) were purchased from Thermo Fisher Scientific. RAW 264.7 cells were obtained from ATCC. MTS (3-(4,5-dimethylthiazol-2-yl)-5-(3-carboxymethoxyphenyl)-2-(4-sulfophenyl)-2H-tetrazolium) assay (Promega) was purchased from Abcam. Tryptic soy agar (TSA) and tryptic soy broth (TSB) were purchased from BD biosciences. HA hydrogel (Belotero Intense) was purchased from Fillerworld. Milli-Q water (18.2 MΩ.cm) was used in all experiments.

### Nanocluster synthesis and optimisation

The protocol of nanocluster synthesis and purification was adapted from published literature[32]. Many parameters for the reaction conditions were adjusted for the nanocluster synthesis to optimise the catalytic activity and retain the ultra-small size. We chose various ratios of metal to ligand with different reducing agents to get ligand-capped nanoclusters (1:0.5, 1:0.8, 1:1,1:1.5 and 1:2) with different solvents in the reaction mixture (Supplementary Table 1). Typically, freshly prepared HAuCl$_4$ or H$_2$PtCl$_6$ (20 mM, 50 μL) was added into DI water, which was followed by quick addition of the reducing agent GSH (5 mM, 200 to 400 μL depending on the metal-to-GSH ratios) as weak reducing conditions, and NaBH$_4$ (143 mM, 20 μL) on top of GSH added for capping as the strong reducing condition. The whole reaction volume was topped up to 1 mL with H$_2$O. Subsequently, the weak reducing condition mixture was heated up to 70 °C and reacted for 24 h, while the mixture under strong reducing conditions was kept at room temperature in thermomixer at a gentle shaking speed of 500 r.p.m. and reacted for 2–3 h. Different reaction temperatures under weak reducing conditions were evaluated for the synthesis, varying from 37 °C, 50 °C, 70 °C to 90 °C. The pH of the reaction under weak reducing condition was varied from 2 to 10. After the 24-h reduction reaction, nanoclusters were purified and buffer exchanged with DPBS (pH 7.4) using Amicon Ultra-15 Centrifugal Filter Units (MWCO 10 kDa) at 5000 × $g$ and 15 min for three times. Finally, the AuNCs in the concentrate were suspended in DPBS to yield 20 μM (concentration given based on a previous study[32]) and subsequently transferred and filtered into a new Eppendorf tube after passing through a sterile syringe filter (0.22 μm Millex-GV Filter) to remove any aggregates and sterilise the sample.

### Thrombin sensor assembly and thrombin cleavage assay

Here, thrombin-cleavable biotinylated peptide ligands (sequence of biotin-SGGfPRSGGSGGC from a previous study[32]) were used in AuNC synthesis for the subsequent conjugation to NAv forming a covalent sensor. For thrombin sensor pH 2, thrombin peptide templated AuNCs pH 2 were synthesised by mixing HAuCl$_4$ (20 mM, 100 μL), DI water (750 μL) and GSH (20 mM, 125 μL) with addition of thrombin cleavable peptide (20 mM, 25 μL), reacting at 70 °C, 500 r.p.m. for 24 h. After purification of the AuNC-thrombin peptide (1 mL), it was mixed with NAv (8 mg/mL, 125 μL) and reacted at 37 °C, at 500 r.p.m. for 3 h. The AuNC-NAv thrombin sensor was then purified using Amicon Ultra-15 Centrifugal Filter Units (MWCO 30 kDa) at 5000 × $g$ and 15 min for at least five times. Finally, the thrombin sensor AuNCs pH 2 were suspended in DPBS to 1 mL (final 1 mg/mL in terms of NAv). For thrombin sensor pH 10, crude AuNC-GSH pH 10 was synthesised by mixing HAuCl$_4$ (20 mM, 100 μL), DI water (736 μL) and GSH (20 mM, 150 μL) with addition of NaOH (1 M, 14 μL), reacting at 70 °C, 500 r.p.m. for 24 h. Thrombin cleavable peptide (20 mM, 25 μL) was added into crude AuNC-GSH pH 10 for another 12-h reaction at 70 °C and 500 r.p.m. The conjugation of AuNC-peptide on NAv, purification, and final stock preparation in DPBS (1 mg/mL in terms of NAv) was the same as for the thrombin sensor pH 2. For the cleavage assay, the purified thrombin sensors pH 2 and 10 (80 μL, 1 mg/mL) were incubated with thrombin enzyme (80 μL, 100 nM) for 3 and 16 h at 37 °C under shaking condition (300 r.p.m.). The incubated solutions were then centrifuged through

Amicon Ultra-0.5 Centrifugal Filter Units (MWCO 30 kDa) at 14,000 × $g$ for 15 min to collect the filtrates containing liberated AuNCs for TMB assay. TMB asasy was conducted by mixing 10 μL of filtrates with 90 μL TMB:H$_2$O$_2$ (2 M) at 1:1 ratio, follow by absorbance measurement at 652 nm using a SpectraMax M5 multimodal microplate reader.

### Liposome preparation and optimisation

To optimise the responsiveness of the liposome system, different lipid compositions were used, including BSM, PC, CH and DOTAP. 1 mol % DSPE-PEG2K was added in all the formulations for prevention of liposome aggregation. The optimised liposome composition was 33.3% BSM, 33.3% PC and 33.3% CH (mass percentage) with 1 mol % of DSPE-PEG2K (all the other compositions tested are summarised in Supplementary Table 3). The lipid films (typically 5 mg total) were prepared in 1.5-mL glass vials through evaporation of CHCl$_3$:MeOH (9:1 used to prepare the lipid stocks) by drying with N$_2$ gas, then storing in a desiccant overnight. The lipid films were hydrated with AuNCs cargo solutions in PBS (500 μL, 2 mM, 100X concentrated compared to above mentioned AuNCs 20 μM as 1X) for at least 1 h. Subsequently, liposomes were formed by vortexing and five freeze-thaw cycles by heating up to 50 °C and freezing at -80 °C. The lipid suspensions were then extruded by using 200 nm polycarbonate membranes (6 cycles) and 100 nm membranes (21 cycles). Finally, the extruded liposome suspension was purified by using size exclusion chromatography (SEC), through a 30-cm column packed with Sepharose CL-2B equilibrated in PBS. Eluted fractions (30 mL) were collected for analysis of the liposome fractions and free AuNC fractions, respectively. Amicon Ultra-15 Centrifugal Filter Units (MWCO 100 kDa) were used to concentrate the eluted liposome fractions. The samples were finally sterilised by filtration through a sterile syringe filter (0.45 μm Millex-GV Filter) to remove any aggregates and sterilise the sample.

### Bacterial culture and toxin supernatant collection

TSB and M-17 broth (3 w/v %) was prepared and autoclaved for growing *S. aureus* SH1000 and *L. lactis* NZ9800 strains, respectively. Both *S. aureus* and *L. lactis* bacterial stocks were initially incubated in TSB or M-17 broth (5 mL each) overnight (12–16 h). The overnight *S. aureus* culture was diluted 100 and 1000 times, then incubated overnight for different time points (ranging from 0 to 24 h). Samples from specific time points were taken for plate streaking and CFU counting to determine the growth curve. For toxin collection, both first overnight stock cultures were centrifuge at 3,500 g for 10 min at 4 °C, followed by sterile filtration of the toxin supernatants through a 0.45 μm Millex-GV syringe filter to remove any remaining bacteria for later incubation of the supernatants with the sensor.

### In vitro incubation of liposomal sensor with toxin, media and thrombin for specificity assay

The prepared AuNC-loaded liposomes (90 μL, 12.5 mg/mL) were incubated with *S. aureus* and *L. lactis* culture supernatants from overnight cultures (90 μL each), TSB (90 μL, 3 w/v %), M-17 broth (90 μL, 3 w/v %), and thrombin (90 μL, 100 nM), respectively, for 3 h at 37 °C under shaking condition (180 r.p.m.). Subsequently, all the samples were filtered through a syringe filter (0.45 μm Millex-GV Filter) and then centrifuged through Amicon Ultra-0.5 Centrifugal Filter Units (MWCO 100 kDa). The filtrates were collected for TMB assays (20 μL of filtrates were added with 180 μL of TMB and H$_2$O$_2$ (2 M) in a 1:1 ratio), and the absorbance was measured at 652 nm using a SpectraMax M5 multimodal microplate reader.

### Stability test of liposomal sensor in TSB

AuNC-loaded liposomes (0.5 mL, 20 mg/mL) were incubated in TSB (4.5 mL, 3 w/v %) ranging from 0 to 48 h. At each time point, 500 μL incubation solution was taken for syringe filtration (0.45 μm Millex-GV Filter) and then passed through Amicon Ultra-0.5 Centrifugal Filter

Units (MWCO 100 kDa). Filtrates were collected for the subsequent TMB assay to determine the stability of the liposomal sensors. TMB assays were conducted mixing 100 μL of filtrate and 100 μL of TMB/$H_2O_2$ (2 M) at a ratio of 1:1 and finally measuring the absorbance at 652 nm.

## Characterisation of NCs and liposomes

The hydrodynamic diameter of nanoclusters and liposomes was measured by using dynamic light scattering (DLS, Zeta Sizer Nanoseries, Malvern Instruments, Ltd) SpectraMax M5 multimodal microplate reader was used to measure the kinetics reaction of TMB oxidation by AuNC catalysts as well as AuNC-loaded liposomes in the presence of $H_2O_2$. The TMB assay was employed to compare the catalytic activity, using a typical condition of a commercial TMB substrate adjusted to 1 M $H_2O_2$ and pH 3, unless specified otherwise. The catalytic activity was quantified by calculating the onset velocity, analysing the kinetics of the absorbance over time, derived from the linear portion of the curve at the onset of the reaction. Transmission electron microscopy (TEM) images of AuNCs were obtained on a JEOL 2100 F. To prepare AuNC samples for TEM, AuNCs solution needed to be desalted first with Zeba Spin Desalting Columns (7 K MWCO) and 5 μL of the sample was dropped on a carbon-coated copper grid and let to dry overnight for imaging. The calibration curve of liposomal sensor was done using TMB oxidation assay. 25 μL of diluted liposomal sensor was mixed with 100 μL of 1-step TMB substrate and 100 μL $H_2O_2$ (2 M). Absorbance at 652 nm of the mixture was measured on the plate reader. The onset velocity was calculated by the increased absorbance signal over time, representing the catalytic activity determined. The minimum detectable sensor threshold was determined as the mean value of the blank measurements plus three times the standard deviation. Matrix-assisted laser desorption/ionisation time-of-flight (MALDI-TOF) mass spectrometry analyses were carried out using a Shimadzu MALDI-8030 mass spectrometer (Shimadzu, UK). MALDI-8030 was operated in linear negative mode. The TOF was calibrated with PEG300 prior experiments. The laser was 200 Hz solid-state, 355 nm. Laser power was 100 and ion gate blanking 500. Mass range was m/z 500 to 15,000. Data were acquired at various ion gate blanking values of 500, 2000, 5000 and 10,000. This allows to filter out ions below a certain mass range. The pulse extraction was set at 5,000 or 10,000 to acquire data. Approximately 10 ng of freeze-dried AuNCs were loaded on the MALDI plate. Nanoparticle Tracking Analysis (NTA) was used to obtain the hydrodynamic diameters, size distribution and count-based concentration of liposomes. For Cryo-Transmission electron microscopy (cryo-TEM), 5 μL of 5 mg/mL liposome solution was dropped on a holey carbon-coated copper grid and subsequently blotted with Whatman 1 filter paper, followed by vitrification into liquid ethane at -180 °C. Frozen grids were placed on a Gatan Multi-Specimen cryo-holder Model 910 and transferred into a Talos L120C Electron microscope. Micrographs were recorded using 4 K x 4 K Ceta CMOS camera, at an accelerating voltage of 120 kV at -175 °C using a low-dose system (40 e-/Å$^2$).

## Molecular dynamics characterisation of NCs

All-atom MD simulations in explicit aqueous solvent were performed using the GROMACS 2022 software package[84]. Glutathione peptide interatomic interactions were modelled via the Amber99sb-ildn FF[85], with additional parameters optimised for N-terminal γ-glutamyl and protonated C-terminal glycine moieties. The TIP3P model was used for water[86], with optimised monovalent Na$^+$ and Cl$^-$ nonbonded parameters from Joung et al.[87]. The PyRED server[88,89] was employed for atom type assignment and restrained electrostatic potential (RESP) partial atomic charge fitting[90] (Supplementary Fig. 6a). Quantum mechanical geometry optimisations and Merz−Singh−Kollman ESP grid fitting[91,92] were performed using Gaussian 16[93] at the HF/6-31 G* level of theory, ensuring compatibility with the Amber FF. The Au$_{25}$

nanoclusters were described using the thiolate-protected gold nanocluster parameters of Pohjolainen et al.[50]. To address the sampling of unphysical tilting and folding of [−Au−S−Au−S−Au−] "staple" geometries in the Au$_{25}$ FF[51], we introduced additional parameters for the 24 angles defined by the harmonic potentials (equilibrium angle of 125° and force constant of 500 kJ/mol) to effectively limit the distortion of the staple motifs (Supplementary Fig. 6b).

Models for Au$_{25}$(SG)$_{18}$ were constructed using a series of preliminary MD simulations and the NanoModeler webserver[94]. First, 50 nanoseconds of MD (at 298.15 K) was conducted on single molecules of protonated glutathione (GSH$^0$) and deprotonated glutathione (GSH$^−$) in aqueous solution. The most compacted peptide conformations for each protonation were extracted and inputted as templates to NanoModeler for the 18 ligands on Au$_{25}$. This was necessary to avoid steric clashes when generating the passivated NC models. Next, 100 nanoseconds of MD was performed at elevated temperature (600 K) for Au$_{25}$(SG$^0$)$_{18}$ and Au$_{25}$(SG$^−$)$_{18}$ in aqueous solvent, with the cartesian coordinates for sulphur and gold atoms frozen during the simulations (Supplementary Fig. 6c). The high temperature allowed for an ensemble of ligand configurations to be quickly sampled while retaining the underlying Au$_{25}$S$_{18}$ crystal structure framework[95]. Cluster analysis and 2D pairwise RMSD similarity matrices (Supplementary Fig. 6d−f) were used to identify ten unique Au$_{25}$(SG)$_{18}$ structures with dissimilar ligand arrangements (Supplementary Fig. 6g). Each of the ten Au$_{25}$(SG$^0$)$_{18}$ and ten Au$_{25}$(SG$^−$)$_{18}$ unique structures were used as the inputs for MD to explore the effect of glutathione protonation and NaCl salt concentration on the conformation of aqueous Au$_{25}$(SG)$_{18}$. Nanoclusters were solvated in periodic cubic boxes (side length ~6 nm) with approximately 5300–7400 water molecules (density ~1 g/cm$^3$). Sodium and chloride ions were included to neutralise each system with additional ions added for the systems with 0.2 M and 1 M NaCl concentrations.

MD simulation parameters were as follows. Electrostatics were evaluated using the Particle Mesh Ewald (PME) method[95] with a real space cutoff of 1.2 nm and a 0.12 nm fast Fourier transform (FFT) grid spacing. Van der Waals interactions were smoothly force-switched to zero from 1.0–1.2 nm. Energy minimisation was carried out using the steepest descent algorithm. MD was performed in the isothermal−isobaric NPT ensemble (298.15 K and 1 atm) using the Nosé-Hoover thermostat[96,97] and Parrinello-Rahman barostat[98]. The LINCS algorithm[99] was applied to constrain all bonds to their equilibrium lengths and improve performance, enabling an integration timestep of 2 fs to be used throughout. Simulations were conducted for 500 nanoseconds each with frames outputted to trajectory files every 5 picoseconds. Each unique structure was simulated twice, with different randomised initial atomic velocities. In total, each of the 6 systems (ligand protonation and salt concentration) was simulated with 20 independent, 500 nanoseconds trajectories. Unless otherwise stated, statistical analysis was performed on the final 300 nanoseconds of each simulation.

## ICP-MS on AuNC-loaded liposomes

Aqua regia was freshly prepared with a molar ratio of 3:1 of HCl (35%, 8 mL) to HNO$_3$ (65%, 2 mL). Liposomes loaded with varying concentrations of AuNCs, ranging from 12.5X (0.25 mM) to 200X (4 mM), or initial AuNC batches for loading (200 μL), were digested in 500 μL of aqua regia overnight and then diluted with $H_2O$ up to 5 mL. A gold standard for ICP was prepared to calibrate the instrument. The gold content in the digested and diluted samples was measured using ICP-MS (8900 Triple Quadrupole ICP-MS), and the data were recorded on iTEVA iCAP Software (Version 2.2.0.51).

## Fluorescence correlation spectroscopy (FCS)

FCS using fluorescently labelled AuNCs was performed similarly to our previous report[32]. Briefly, AuNCs (1 mM, 50 μL) were labelled through

reactive amine (GSH on AuNCs) coupling by adding a 20-molar excess of NHS-activated Oregon Green 488 (OG488). This labelling process lasted for 3-h incubation at room temperature. Subsequent PD Mini-Trap and PD MidiTrap columns equilibrated in PBS were then utilised to remove any unreacted OG488 dye to obtain purified AuNCs-OG488. Subsequently, 5 mol % labelled AuNCs-OG488 were mixed with 95 mol % unlabelled AuNCs (overall final conc. 2 mM) and encapsulated in BPC liposomes, which were again purified by SEC using a Sepharose CL-2B column. The purified AuNC-liposomes were then incubated 1:1 with either TSB or *S. aureus* SH1000 overnight culture supernatant containing the secreted toxins in TSB and incubated for 2 h, at 37 °C. FCS was then performed on a commercial LSM 880 (Carl Zeiss, Jena, Germany) using 5 μL droplets on glass-bottom ibidi 8-well plates (80827, ibidi, Germany), equilibrated at 37 °C. The setup included an Ar$^+$ laser (488 nm excitation), 40X C-Apochromat water immersion objective (NA 1.2), and a LP 505 filter. All measurements were performed 200 μm above the ibidi bottom plate. A serial dilution of OG488 diluted in PBS or TSB were used for calibration ($D = 5.49 \times 10^{-6}$ cm$^2$/s at 37 °C)[100] and obtaining the x-y dimension of the confocal volume ($\omega^2_{xy}$) to enable calculation of diffusion coefficients ($D$) and use Stokes-Einstein equation to convert to hydrodynamic diameter ($D_h$). Intensity traces ($25 \times 5$ s) were recorded and analysed for each sample. The autocorrelation curves shown in the figures represent the average curve for the whole 125 s measurement. All curves were fitted and analysed using PyCorrfit programme 1.1.6[101]. Free AuNCs-OG 488 were first fitted with a one component fit to obtain their characteristic diffusion time and the liposome samples were then analysed with a two-component fit, fixing one component to free cluster diffusion. The ratio of height to width of the confocal volume was fixed to 5 and a triplet fraction with triplet time of 1-10 μs was included in all fits. The number of clusters loaded per liposome were obtained by using the counts per particle (cpp in kHz) signal of free AuNCs-OG488 and loaded liposomes, using a correction due to the disproportional effect of bright slow diffusing species on average autocorrelation curves[59], and multiplying by 20 because we only used 5 mol % of loaded AuNCs in the encapsulation procedure.

**Single particle automated Raman trapping analysis (SPARTA®)**
SPARTA®, as previously reported[60], is a label-free method that enables high-throughput analysis of single nanoparticles with Raman spectroscopy to provide population level information with single particle detail about nanoparticle size and chemical composition. For sample preparation, the optimal BPC liposomes were prepared and loaded with catalytically optimised AuNCs through lipid film hydration and extrusion. To collect fresh toxin supernatant, *S. aureus* SH1000 strain was cultured in TSB overnight for bacterial growth and toxin secretion. For sample incubation, BPC liposomes (10 mg/mL, 200 μL) were incubated with DPBS (200 μL) as control or 4-fold diluted fresh toxin supernatant in DPBS (200 μL) for 2 h. SPARTA® measurements were conducted using the SPARTA® Alpha prototype. A 22 mm diameter coverglass with thickness 0 (VWR) was affixed to a microscope slide. Onto the coverglass was placed 200 μL of either particle suspension or DPBS for background measurement. During measurements, liposomes in DPBS or with toxin were trapped and analysed with a 10 s acquisition time. The laser was disabled for 1 s to release the trapped particle and allow a new particle to diffuse into the confocal volume. 20 spectra of blank DPBS at 10 s acquisition time was measured for background subtraction. Raman spectra were analysed using custom MATLAB scripts for cosmic spike removal and spectral response correction (785 nm reference standard National Institute of Standards and Technology, US). Aggregates or empty traps were then manually removed by minimum and maximum thresholding, before a background subtraction of 95% blank DPBS was performed. A Whittaker baseline correction was then applied, before each spectrum was then smoothed using a Savitzky-Golay smoothing filter with order 1 and frame size 7.

Each particle spectrum was normalised to the area under the whole spectrum to remove effects such as laser power fluctuation between samples. Mean spectra and standard deviation were calculated across each sample in Matlab R2020a.

**In vitro toxin-triggered release of AuNC from liposome**
Toxin was collected from an overnight culture supernatant of *S. aureus* SH1000 cultured in TSB, centrifuged away the bacteria, and sterile filtered the supernatant before use. Different formulations of liposomes (5 mg/mL, 250 μL) were mixed with the toxin supernatant (250 μL) and incubated at 37 °C for 2 h at a shaking speed of 500 r.p.m. As a control group without toxin, and a positive full release control, liposomes were separately incubated with the same volume of TSB and Triton-X100 in PBS (10 v/v %), respectively. After incubation, the reaction solutions were centrifuged and filtered through Amicon Ultra-0.5 Centrifugal Filter Units (MWCO 100 kDa) to separate the released AuNCs from the intact liposomes. The collected filtrates were collected for TMB assay for detection of released AuNCs. To perform the TMB assay, all the collected filtrates (100 μL) were added into a 96-well plate, followed by the addition of 1-Step™ Ultra TMB-ELISA Substrate Solution (50 μL) and H$_2$O$_2$ (2 M, 50 μL). The kinetics of the reactions were measured by monitoring the absorbance at a wavelength of 652 nm using a SpectraMax M5 plate reader.

**In vitro AuNC release from bacterial incubation with liposome**
*S. aureus* SH1000 was cultured overnight to obtain 1X stock which was subsequently diluted at a 1:100 ratio. BPC liposomes were prepared at a concentration of 20 mg/mL in a total volume of 2 mL. The prepared liposomes (1 mL) were incubated with 5 mL of the diluted *S. aureus* SH1000 in TSB (1:100) or 5 mL TSB only was used as the control. The incubation period lasted for 8 h. At various time points (0–8 h) with 1-h intervals, 500 μL of the reaction mixture solution (3.33 mg/mL of lipids) was sampled. Each sample was then centrifuged and filtered using Amicon Ultra-0.5 Centrifugal Filter Units (MWCO 100 k Da). All the collected filtrates (100 μL), which contained the released AuNCs, were used for the TMB assay (60 μL 1-Step TMB and 40 μL H$_2$O$_2$). This assay allows for the quantification of the released AuNCs by measuring oxidised TMB absorbance.

**In vitro tests with HA hydrogels**
BPC liposomes loaded with AuNCs were prepared at a concentration of 20 mg/mL. *S. aureus* SH1000 was cultured in TSB overnight and subsequently purified by pelleting and resuspension in DPBS. The purified SH1000 was then diluted to a concentration of 10$^8$ colony-forming units per millilitre (CFU/mL) in DPBS. 200 μL of BPC liposomal sensors were added to 200 μL of HA hydrogel (Belotero Intense). Subsequently, either TSB (200 μL, 3 w/v %) or the diluted SH1000 (200 μL, 10$^8$ CFU/mL) was added to the above mixture (BPC liposomal sensors with HA hydrogel). After thorough mixing, the resulting mixtures were pipetted and injected into 1400 μL of TSB for incubation over a period of 0–48 h. To analyse the release of AuNCs, 200 μL of supernatant was collected from each incubation solution at different incubation time points and filtered using a 0.45 μm syringe filter into an Amicon Ultra-0.5 mL Filter Unit (100 kDa). This filtration step allowed for the separation of the released AuNCs into the filtrate, which was then used for the subsequent TMB assay. For the TMB assay, 20 μL of the filtrate was mixed with 180 μL of TMB:H$_2$O$_2$ solution (final concentration 1 M H$_2$O$_2$) for the generation of a colourimetric signal. The absorbance of the reaction mixture was measured over time on the plate reader.

**In vitro cytotoxicity studies**
RAW 264.7 cells were cultured in culture media consisting of DMEM 1X medium with 10% of FBS and 0.5% of penicillin-streptomycin. To seed the cells, the desired concentration of RAW 264.7 cells were prepared

in culture media and then transferred into sterile 96-well plates (20,000 cells/well), which is followed by incubation for 24 h for adhesion. The culture media from the wells with the adherent RAW 264.7 cells was removed and different concentration of AuNC-loaded liposomes and AuNCs (20 μL both in PBS) were added in the wells with a final mixture of 180 μL of medium:PBS (9:1). The plates were incubated at 37 °C with 5% $CO_2$ for another 24 h. The cell viability assay was evaluated by using reagents MTS assay (MTS) by removing the cell supernatants and replacing with 120 μL of MTS stock solution containing DMEM medium without phenol red (31.5 mL), MTS (2 mg/mL, 6 mL), PMS (0.92 mg/mL, 0.3 mL), incubation for 1 h and measurement of absorbance at 490 nm.

### In vivo injection and urine analysis

All animal experiments were conducted according to the Animals (Scientific Procedures) Act 1986, following ethical review by Imperial College London Animal Welfare and Ethical Review Body and with an internal ethics board and UK government approved project (PF93C158E and PP5168779) and personal license (I87985646; Peeler), and with approval from the UK Home Office authority; studies followed ARRIVE guidelines. For AuNC clearance study, the catalytically optimised AuNCs (10 μM, 200 μL) or DPBS (200 μL) were injected i.p. into each mouse. Urine samples were collected in Eppendorf tubes after 1-h and 3-h period post injection for TMB assay and then diluted to get enough volume for the assay. The diluted urine sample (10 μL) was mixed with 40 μL of TMB:$H_2O_2$ solution (final $H_2O_2$ concentration of 2.5 M) in 384-well plate for the absorbance measurement at 652 nm. The results were normalised by dilution factors for comparison. This dilution was necessary due to the limited volume of urine collected. A mixture of liposomes (60 mg/mL, 100 μL), hyaluronic acid (HA) (100 μL), and either DPBS or *S. aureus* SH1000 ($3 \times 10^8$ CFU/mL, 100 μL) was prepared at room temperature. Subsequently, 200 μL of the above prepared mixture with PBS (healthy control, HC) or SH1000 (bacterial infection, BI) was injected i.p. into each mouse (wild-type C57BL/6 mice purchased from Charles River, female, 6–14 weeks old). Adult mice were housed no more than five per cage with Aspen chip 2 bedding with a 12-h light and 12-h dark cycle at 20–22 °C. Mice were randomly assigned to experimental groups. Water was provided freely, and the mice were fed RM1 (Special Diet Services). Groups consisted of 6 animals each. After a 4-h period after injection, urine samples were collected in Eppendorf tubes for further analysis. The collected urine (20 μL) was subjected to the TMB assay in a 384-well plate, where it was mixed with 60 μL of TMB:$H_2O_2$ solution (final concentration of 5 M). Mice were humanely sacrificed via cervical dislocation and death was confirmed by severing the femoral artery.

### Reporting summary

Further information on research design is available in the Nature Portfolio Reporting Summary linked to this article.

## Data availability

All data supporting the findings of this study are available within the article and its supplementary files. Any additional requests for information can be directed to, and will be fulfilled by, the corresponding authors. Source data are provided with this paper.

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

## Acknowledgements

We kindly acknowledge Dr Akemi Nogiwa Valdez for editing of the manuscript and data management support. A.N. acknowledges support from a Sir Henry Wellcome Postdoctoral Fellowship (209121_Z_17_Z) from the Wellcome Trust. A.K. acknowledges from the European Union's Horizon 2020 research and innovation programme under the Marie Skłodowska-Curie Actions grant agreement "BacDrug" [838183]. C.S. acknowledges funding from the EPSRC Centre for Doctoral Training in the Advanced Characterisation of Materials (EP/S023259/1). L.P. and M.M.S. acknowledge support from Cancer Research UK (EDDPMA-May23/100017). C.T. acknowledges support via the Office of the Civil

Service Commission (OCSC) - Royal Thai Government Scholarship scheme. P.C. and I.Y. acknowledge the high-performance computing resources provided by the Australian Government through the National Computational Merit Allocation Scheme (NCMAS project e87). I.Y. and M.M.S. acknowledge funding from the Australian Research Council under the Discovery Project scheme (Grants No. DP170100511 and DP230100709). I.S. acknowledges funding from the Fundação para a Ciência e Tecnologia (2022.13654.BD). We thank the Light Microscopy Facilities at the Francis Crick Institute (London, UK) for access to FCS, and characterisation facilities within the Harvey Flower Electron Microscopy Suite, Department of Materials, Imperial College London. This research was funded in part by the Wellcome Trust (209121_Z_17_Z). M.P. and M.M.S. acknowledge funding from the Engineering and Physical Sciences Research Council for M.P.'s studentship (EP/L015498/1) through the Institute of Chemical Biology at Imperial College London. M.M.S. acknowledges the EPSRC IRC Agile Early Warning Sensing Systems for Infectious Diseases and Antimicrobial Resistance (EP/R00529X/1), the Rosetrees Trust and the Department of Science, Innovation and Technology (DSIT) and the Royal Academy of Engineering under the Chair in Emerging Technologies programme (CiET2021\94). For the purpose of open access, the author has applied a CC BY public copyright license to any Author Accepted Manuscript version arising from this submission.

## Author contributions

K.C., A.N., and M.M.S. conceived the project and wrote the paper. K.C. and A.N. primarily designed the experiments. K.C. executed the majority of experimental procedures and data processing. P.C. and I.Y. carried out molecular dynamics simulations. C.S. conducted SPARTA® measurements. A.N. performed FCS measurements. C.T. conducted TEM imaging of AuNCs and M.C. performed cryo-TEM imaging of liposomes. A.K. and A.M.E. assisted with bacterial culture. I. S. helped with AuNC synthesis. L.P., K.K. and I.P.P helped MALDI-TOF measurements. M.P. contributed to the help for liposome assembly. T.B.C. D.J.P. and J. S. T. conducted AuNC and liposomal sensor administration into mice. C.N.L. and L.C.F helped experimental design and manuscript proofreading. All authors engaged in result discussions and manuscript preparation. M.M.S. supervised the project and assisted in the study design.

## Competing interests

The authors declare the following competing financial interest(s): M.M.S. has filed a patent application (1810010.7) and has a registered trademark (US Reg. No. 6088213) covering the name SPARTA® and the techniques, described in the manuscript by Penders et al. https://doi.org/10.1038/s41467-018-06397-6. M.M.S. is a founder of Sparta Biodiscovery Ltd. C.N.L. and M.M.S. have filed a patent application (US20200116725 A1) on renal clearable nanocatalysts for disease monitoring. The remaining authors declare no competing interests.

## Additional information

¹Department of Materials, Department of Bioengineering, and Institute of Biomedical Engineering, Imperial College London, London SW7 2AZ, UK. ²School of Cancer & Pharmaceutical Sciences, Institute of Pharmaceutical Science, King's College London, London SE1 9NH, UK. ³School of Engineering, RMIT University, Melbourne, VIC 3001, Australia. ⁴Centre for Bacterial Resistance Biology (CBRB), Department of Infectious Disease, Imperial College London, London SW7 2AZ, UK. ⁵Department of Infectious Disease, Imperial College London, London SW7 2AZ, UK. ⁶BioEM lab, Biozentrum, University of Basel, Mattenstrasse 26, Basel 4058, Switzerland. ⁷Department of Physiology, Anatomy and Genetics, Department of Engineering Science, Kavli Institute for Nanoscience Discovery, University of Oxford, Oxford OX1 3QU, UK. ⁸Department of Medical Biochemistry and Biophysics, Karolinska Institutet, Stockholm 171 77, Sweden. ⁹Department of Chemistry, University College London, London WC1H 0AJ, UK. ✉e-mail: adrian.najer@kcl.ac.uk; irene.yarovsky@rmit.edu.au; molly.stevens@dpag.ox.ac.uk

