## [Transparent Peer Review file · Nature Communications]

Non-invasive in vivo sensing of bacterial implant infection using catalytically-optimised gold nanocluster-loaded liposomes for urinary readout

Corresponding Author: Professor Molly Stevens

Version 0:

Reviewer comments:

Reviewer #1

(Remarks to the Author)

Authors report very interesting work on using the AuNc nanoclusters with the catalytic properties for the detection of the bacterial (*S. aureus*) infections at the implants. Authors support the experimental findings with the all-atom molecular dynamics (MD) simulations to show the ligands protection of the cluster stability. Finally, the biosensor effect is demonstrated in the release of the AuNC, once in contact with the bacterial toxin and its subsequent renal clearance, using animal model.

As it is described, the work demonstrates rather the incremental findings, compared to their previous famous work on renal clearable catalytic gold nanoclusters for in vivo disease monitoring (Nature Nanotech. 2019). Can authors better explain the originality of this specific work, beyond the selected application?

Another concern is the hydrogel stability itself in the in vivo conditions and its ability to keep the liposomes intact. Is 6 hours sufficient to prove the stability of the 'implant' construct, taking into account that the covalent immobilization between the sensor and implant was not preferred. Authors should try to use the covalent attachment to demonstrate that the concept still works (risk of failure, once no/or significantly reduced release).

I recommend the major revision of the article and additional experiments done.

Reviewer #2

(Remarks to the Author)

This study showed a clear aim of developing and optimizing a GSH-AuNC based bacterial sensing system via liposomal encapsulation. Leveraging ultrasmall AuNC's renal excretion, a colourimetric assay was further employed to be the indicator of bacterial infection.

1. "The peroxidase-like catalytic activity of AuNCs was evaluated by monitoring the absorbance at 652 nm, which resulted from the oxidation of TMB by H₂O₂ catalysed by AuNCs₄₁, which is the assay we used to compare catalytic activity. The typical assay conditions included a commercial TMB substrate adjusted to 1 M H₂O₂ and pH 3, if not specified differently. The assessment involved calculating the onset velocity by analysing the kinetics of the absorbance over time, derived from the linear portion of the curve at the onset of the reaction." This part reads a bit like a patent instead of an academic manuscript, please consider modifying the wording.

2. The GSH-AuNC optimization results are interesting, and the authors discussed heavily on the underlying mechanism to explain the fact that "AuNCs exhibited a similar molecular weight, ranging from 2,000 to 12,000 Da, along with an identical Au-to-GSH ratio..., but significantly different catalytic activity ?" However, this part feels a bit strayed away from the introduction and the aim of bacterial biosensing.

3. The reviewer is sure that two technical repeats, such as the case of Fig. 5B, can show the trend of group differences, but is not sure that one or two repeats are enough to reach a statistical conclusion.

4. Stability (in vitro and/or in vivo) of this liposomal AuNC system should be comprehensively tested.

5. "AuNC-liposomes mixed with HA hydrogel were either bacteria-free or intentionally contaminated with *S. aureus*." Is the use of hydrogel a good simulation of patients with implant infection in real clinical settings? In addition, there is no validation of the (liner or other) responsiveness of the sensing system with the severity of *S. aureus* local infection. When considering *in vivo* stability and bacterial specificity of this sensing system (or the lack of them), the feasibility seems to be in heavy doubt.

6. Would the urine volume, urine color, or collection timing, affect the results? Would other diseases disturb the liposome and release the GSH-AuNC?

7. In hypothetical clinical setting, how would this system being used? It is interesting to read that the authors proposed "a covalent coupling strategy to anchor the sensor to the implant to extend the duration of liposome retention within the hydrogel", "but (the authors) envisage this as future work".

8. The reviewer agrees that the authors modified this sensing system extensively with regards to its catalytic activity, mode of action, and underlying mechanism. But it seems that the results focus more on mechanical exploration other than echoing the introduction and the clinical question.

Version 1:

Reviewer comments:

Reviewer #1

(Remarks to the Author)

Reviewer thanks for the additional work and all experiments done by the authors to improve their manuscript, and particularly their focus to highlight the novelty compared to Nature Nanotech paper. According to their arguments, Reviewer concludes:

1. Methodology of detection stays the same as in the previous work (renal clearance based colorimetric detection of AuNC);
2. Still, catalytic activity of AuNC was notably improved that enables to improve the limit of detection of the bioassay (note, also other works on enhancement of AuNC catalytic activities to improve biosensor sensitivity: <https://doi.org/10.1016/j.microc.2023.108920m>, <https://doi.org/10.1016/j.microc.2023.108920>, etc.). Improved sensitivity was clearly demonstrated using thrombin binding assay (Figure S4), similar to one, also shown in the previous article. Analyte was selected to highlight the enhancement of the catalytic activity of AuNC;
3. Compared to the previous work, authors focused on different application and validated it using murine model (bacterial infections in the implants);
4. Authors used noncovalent binding of liposomes, compared to covalent protein-gold conjugates before that overall should make the biosensor design more flexible and universal.

(Related to the point 4: authors did very extensive works to test the stability of the hydrogel-GoldNC construct over the longer time and in differently conditioned media (Point 2 of the previous report). The only concern that just came into Reviewers mind is that the specificity studies could be done to make sure that no other similar bacterial infections or other factors, e.g. proteases will not break the binding and this will not lead to the false positive result).

Reviewer #2

(Remarks to the Author)

The authors have adequately addressed all my concerns, by presenting more experimental results and clarifying potential misleading expressions. In addition, the significance and novelty of this work, in comparison with previous reports, have been demonstrated clearly in this revised manuscript.

Point-by-point response to the reviewers

We kindly thank the reviewers for their constructive feedback, which we have thoroughly addressed. The reviewers' comments are in black, the authors' point-by-point responses are in blue, and the manuscript changes are in red.

Reviewer #1 (Remarks to the Author):

Authors report very interesting work on using the AuNC nanoclusters with the catalytic properties for the detection of the bacterial (*S. aureus*) infections at the implants. Authors support the experimental findings with the all-atom molecular dynamics (MD) simulations to show the ligands protection of the cluster stability. Finally, the biosensor effect is demonstrated in the release of the AuNC, once in contact with the bacterial toxin and its subsequent renal clearance, using animal model.

1. As it is described, the work demonstrates rather the incremental findings, compared to their previous famous work on renal clearable catalytic gold nanoclusters for in vivo disease monitoring (Nature Nanotech. 2019). Can authors better explain the originality of this specific work, beyond the selected application?

Thank you for your insightful comment. We have now updated the manuscript to better highlight the novelty of the current study, which introduces several transformative advancements. First, we enhanced the catalytic activity of AuNCs by a striking 33.5-fold compared to our previous work using synthesis optimisation. This will impact not only the biosensing but also therapeutics fields, where nanocatalysts are being evaluated for anti-cancer and antioxidant treatments. To further demonstrate the impact of the higher AuNC catalytic activity beyond the current application, we have now added additional data using our previous sensor setup as biosensing unit. We synthesised peptide-linked AuNC-protein conjugates for thrombin sensing, comparing sensors made with the previous and the new AuNCs (Supplementary Figure 4g). The sensor with optimised AuNCs yielded 6x higher signal and 10x higher signal-to-noise, confirming broad impact of the higher catalytic activity of AuNCs in improving sensitivity. Second, the current paper describes a completely different biosensor design and application, encapsulating AuNCs in toxin-responsive liposomes for bacteria sensing compared to the covalent protein-AuNC conjugate used for cancer detection in our previous work. The current biosensor design is potentially more versatile and scalable compared to our previous work as the AuNCs are produced without any specific peptide modification and therefore can be more rapidly adapted to a range of applications. Thirdly, we provide extensive mechanistic evaluation of our new sensor design, which includes various single-particle techniques (cryo-TEM, FCS, SPARTA) and computational analysis by atomistic simulations. The text has now also been updated to better present these highlights.

“This improvement in AuNC catalytic performance can enhance the sensitivity of the sensing platform, allowing for detection even with minimal liberation of AuNCs, which is particularly relevant when aiming to detect disease early. Compared to our previous work³², we improved the catalytic activity of AuNCs, changed the sensor design away from a covalent peptide-AuNC conjugate to a non-covalent liposomal system, expanded application from cancer to bacterial infection, and provided extensive mechanistic understanding by leveraging various single-particle techniques and computational analysis.”

“This indicates an 83-fold increase in sensitivity for the herein optimised clusters (Au-to-GSH 1:1.5). The broad impact of AuNCs with higher catalytic activity was demonstrated by formulating a thrombin sensor using peptide-linked AuNC-protein conjugates analogous to our previous study with non-optimised AuNCs³². The sensors developed with the optimised AuNCs synthesised at pH 10 showed 6x higher signal and 10x higher signal-to-noise (S/N) compared to our previous study, confirming the importance of catalytic optimisation on diagnostic sensitivity (Supplementary Figure 4g-i).”

Supplementary Figure 4. Characterisation of AuNC 1:1 and 1:1.5 synthesised at different pH. (a) Photographs of (i) final products of all AuNCs synthesised at different pH yielding different colours and (ii) colourimetric readout after TMB oxidation assay for

all AuNCs. (b) Inductively coupled plasma mass spectrometry (ICP-MS) measurement of gold content of AuNC-GSH 1:5 synthesised at pH 2 and 10 (mean values \pm standard error, $n = 3$ technical repeats). This shows that the gold content from both AuNCs had similar amounts. (c) and (d) UV-vis measurement of both AuNC-GSH 1:1 and 1:1.5 synthesised at different pH reaction conditions. This showed AuNCs synthesised at pH 6 might have produced a lower concentration and could account for the lower catalytic activity ($N = 3$ samples, $n = 1$ measurement). The UV-vis of AuNC-GSH in a 1:1.5 ratio synthesised at pH 2 and 10 exhibited similarities consistent with the gold content measured by ICP-MS (c). (e) and (f) The limit of detection (LoD, calculated as 3 times of standard deviations plus the mean value of the background) of AuNCs 1:1.5 from pH 2 and pH 10 reaction conditions. The catalytic activity was measured with the AuNCs diluted in synthetic urine (50 v/v %) and evaluated by the onset velocity analysis from the oxidation of TMB substrate ($A_{652nm} s^{-1}$). The LoD of AuNCs 1:1.5 from pH 2 and 10 synthesis condition was 2.33 and 0.028 pmol (mean values \pm standard error, $n = 3$ technical repeats), respectively, corresponding to 83-fold enhancement of catalytic activity, with the higher number corresponding to the AuNCs of a related recent paper². (g, h) **Thrombin (THR) cleavage TMB assay on the filtrate after incubation of both THR sensors made of AuNC pH 2 (g) and AuNC pH 10 (h) for 3 and 16 hours of incubation. The signal from THR sensor pH 10 was 6x higher than THR sensor pH 2.** (i) **The signal-to-noise (S/N) ratio (thrombin/PBS) for both sensor for 2 timepoints. THR sensor pH 10 revealed 4.3x and 10.5x higher S/N than THR sensor pH 2.**

2. Another concern is the hydrogel stability itself in the in vivo conditions and its ability to keep the liposomes intact. Is 6 hours sufficient to prove the stability of the 'implant' construct, taking into account that the covalent immobilization between the sensor and implant was not preferred. Authors should try to use the covalent attachment to demonstrate that the concept still works (risk of failure, once no/or significantly reduced release).

I recommend the major revision of the article and additional experiments done.

We appreciate the insightful comment on stability of the liposomal sensor – hydrogel system. We agree that stability beyond the 6 h used in our first proof-of-concept study is desirable and we have now performed and added such experiments:

- Incubating AuNC-loaded liposomes with media (TSB) and *S. aureus* bacteria for up to 48 hours followed by TMB assay on filtrates (Figure 5c-e and Supplementary Figure 19a-f), which revealed release of AuNCs only in presence of bacteria, not in the bacteria-free controls.

- We confirmed liposomal stability by DLS characterisation in different media (PBS, TSB and 10 v/v% human serum) after incubation at body temperature for up to 48 hours (Supplementary Figure 19g-i).
- Incubating HA gels loaded with AuNC-liposomes in media (TSB) with and without *S. aureus* bacteria for up to 48 hours followed by TMB assay on filtrates (Supplementary Figure 22) showed no signal for the controls, whilst bacteria-contaminated gels released the AuNCs.
- Injecting HA gels loaded with AuNC-liposomes in healthy mice and taking urine samples up to 48 hours later yielded no signals, confirming *in vivo* stability of the sensing system in presence of the gel without contamination with pathogenic bacteria (Supplementary Figure 25).

Altogether, these experiments revealed high stability of the sensor system in the absence of pathogenic bacteria *in vitro* and *in vivo*, including when combined with the hydrogel, now tested up to two days each. The rationale of using the commercial HA gel lies in the well-documented stability (up to 12 months in humans)² and current approval and clinical use as dermal fillers. We agree with the reviewer that a covalent immobilisation strategy would be an interesting next step. However, this will require a significant change in the implant and sensor system as well as repetition of the whole manuscript with the new design, which we feel is beyond the scope of the current study. Such a new study will also require significant resources, which can only be conducted upon securing extensive future funding. We have now added the above experimental data, incorporated additional discussion sections, and included the required information and reference on HA gel stability.

“In contrast, the filtrates from liposomes incubated with TSB culture medium showed no colour change even after 48-hour incubation, indicating high stability of liposomes in media in the absence of bacteria (Figure 5c-e and Supplementary Figure 19d-f).”

“The onset velocity of the reaction was consistent with the endpoint measurements, and reproducibility of the assay was confirmed (Figure 5d, e and Supplementary Figure 19a-e).”

“To further verify the stability of the liposomes, DLS characterisation was performed in different media. The results showed that the size of the liposomes remained around 120 nm without disassembly or aggregation after incubation in various media at body temperature over at least 48 hours (PBS, PBS:TSB 1:1, and 10 v/v% human serum) (Supplementary Figure 19f-i).”

“These commercial HA hydrogels offer enhanced stability due to cross-linking technologies, which extend their duration and provide resistance to enzymatic degradation, ensuring a longer-lasting effect in tissue augmentation applications^{75,76}. The

commercial product we used (Belotero intense) is widely used and has a reported longevity of up to 12 months after injection⁷⁷.”

“Incorporating a covalent coupling strategy to anchor the sensor to the implant could extend the duration of liposome retention within/on the implant, but this will require extensive optimisation and we envisage this as future work.”

“Additionally, we administered AuNC-loaded liposomes with HA gel to three healthy mice, along with two PBS injection controls, followed by TMB assays on urine samples collected over a 48-hour period. The absence of blue colour change in both groups demonstrates that the sensing platform remained stable *in vivo* during that timeframe (Supplementary Figure 25).”

Figure 5. *In vitro* incubation of bacteria with AuNC-loaded liposomes and characterisation of bacterial growth. (a) Schematic of *in vitro* incubation of AuNC-loaded liposomes with bacteria and subsequent separation of released AuNCs from intact liposomes. (b) Bacterial growth curve of *S. aureus* SH1000 by CFU counting method (mean values \pm standard error, N = 3 independent repeats). (c-e) TMB oxidation assay on the filtrates after incubation of AuNC-loaded liposome with TSB +/- *S. aureus* from 0 to 24 hours. (c) Photographs of final TMB assay results. (d, e) Absorbance (at 652 nm at 140 second endpoint) and onset velocity for assay in (c): TSB (black bars) and TSB

+ bacteria (grey bars) (mean values \pm standard error, N = 3, n=1, additional repeat with detailed reaction curve in Supplementary Figure 19a-e). (f, g) *In vitro* cytocompatibility MTS cell viability assay after incubation of AuNCs (f) and AuNC-loaded liposomes (g) with RAW 264.7 cells (mean values \pm standard error, N = 3 individual experiments, n = 3 measurements).

Supplementary Figure 19. Response performance of AuNC-loaded liposomes to toxin by co-incubation with *S. aureus* SH1000. (a,b) The measurement of absorbance over time for TMB assay on the filtrates collected after various time periods after incubation of AuNC-loaded liposome with TSB and TSB + bacteria. (c) Absorbance measurement at the endpoint of 140 seconds from (a, b). (d) Onset velocities calculated from (a,b). (e) Photographs show the final TMB assay results from (a-d); Incubation time varies from 0 to 24 hours (N=1). (f) Stability of liposomal sensor in TSB from 0 to 48 hours. The filtrates were collected after incubating liposomal sensor in TSB for 0 to 48 hours, followed by the TMB assay. The insert photograph shows no blue colour change after TMB colourimetric reaction. (g, h, i) DLS measurements of liposomes in PBS (g), PBS:TSB 1:1 (h), and 10 v/v% HS (human serum) (i), after up to 48 hours of incubation.

Supplementary Figure 22. Bacteria-triggered release of AuNCs from liposomal sensors within HA hydrogels. (a) Schematic of the *in vitro* procedure mimicking *in vivo* experiment. (b) TMB assay (absorbance measurement) on filtrates after incubation of the liposomal sensor mixed with HA gel in TSB +/- bacteria, showing absorbance of TMB reaction upon release of AuNCs. The incubation time varied from 0 to 48 hours. (c, d) Release efficiency calculated from onset velocity (Figure 6d) and absorbance (b).

Supplementary Figure 25. *In vivo* stability of liposomal sensor post i.p. administration with HA gel in mice after 0 to 48 hours. Absorbance of TMB assay performed on collected urine post i.p. administration of HA gels with incorporated liposomal sensor into 3 healthy control (HC) mice and PBS administration into 2 mice as negative control (Neg Ctrl). Urine collected after 0 hour (a), 4 hours (b), 8 hours (c), 24 hours (d), and 48 hours (e). (f) Photographs of TMB assay results from (a-e) showing no blue colour change. There were some artifacts, observed as humps (c-e), which were likely associated with the presence of bubbles in the reaction solution after reagent addition. This does not correspond to a TMB signal, as no accompanying blue colour change was observed in (f).

Reviewer #2 (Remarks to the Author):

This study showed a clear aim of developing and optimizing a GSH-AuNC based bacterial sensing system via liposomal encapsulation. Leveraging ultrasmall AuNC's renal excretion, a colourimetric assay was further employed to be the indicator of bacterial infection.

1.“(The peroxidase-like catalytic activity of AuNCs was evaluated by monitoring the absorbance at 652 nm, which resulted from the oxidation of TMB by H₂O₂ catalysed by AuNCs⁴¹, which is the assay we used to compare catalytic activity (see the materials and methods for details). The typical assay conditions included a commercial TMB substrate adjusted to 1 M H₂O₂ and pH 3, if not specified differently. (delete it and move into methods part) The assessment involved calculating the onset velocity by analysing the kinetics of the absorbance over time, derived from the linear portion of the curve at the onset of the reaction.” This part reads a bit like a patent instead of an academic manuscript, please consider modifying the wording.

Thank you for your valuable feedback. We appreciate your suggestion and the wording was revised accordingly. Some of the details were shifted to the method part as highlighted below to ensure the main text aligns more closely with the style of an academic manuscript.

“The peroxidase-like catalytic activity of AuNCs was assessed by measuring the absorbance at 652 nm, which resulted from TMB oxidation via H₂O₂ catalysed by AuNCs⁴⁰, represented throughout as kinetics, endpoint absorbance, or onset velocity.”

Moved to methods part:

“The TMB assay was employed to compare catalytic activity, using a typical condition of a commercial TMB substrate adjusted to 1 M H₂O₂ and pH 3, unless specified otherwise. The catalytic activity was quantified by calculating the onset velocity, analysing the kinetics of the absorbance over time, derived from the linear portion of the curve at the onset of the reaction.”

2.The GSH-AuNC optimization results are interesting, and the authors discussed heavily on the underlying mechanism to explain the fact that “AuNCs exhibited a similar molecular weight, ranging from 2,000 to 12,000 Da, along with an identical Au-to-GSH ratio..., but significantly different catalytic activity?” However, this part feels a bit strayed away from the introduction and the aim of bacterial biosensing.

Thank you for identifying parts of the manuscript that can benefit from a rewrite. We appreciate your interest in the GSH-AuNC optimisation results and the discussion on the underlying mechanism. We acknowledge that this section may appear somewhat divergent from the main focus of bacterial biosensing, which we now aimed to improve

as highlighted below. Optimisation of AuNC catalytic activity, reaching 33.5-fold better signal compared to our previous work³, together with detailed analysis by computational methods are major findings of the current paper (please also see our response to the novelty query from reviewer #1, comment 1). This will have implications not only for bacterial biosensing, but also other sensing platforms using AuNCs for e.g. thrombin sensing (see new Supplementary Figure 4g-i) or more distant fields using catalytic AuNCs for anti-cancer or antioxidant therapy. Our manuscript should now provide a more cohesive narrative after changing the abstract, introduction and aim of the study to better align with the main text. We removed some of the bacteria intro and added more information on AuNCs for sensing and importance of optimising catalytic activity, whilst retaining the same size.

~~“*Staphylococcus aureus* is a leading cause of nosocomial implant-associated infections, causing significant morbidity and mortality, underscoring the need for rapid, non-invasive, and cost-effective diagnostics. Here, we optimised the synthesis of renal-clearable gold nanoclusters (AuNCs) for enhanced catalytic activity with the aim of developing a sensitive colourimetric diagnostic for bacterial infection. All-atom molecular dynamics (MD) simulations confirmed the stability of glutathione-coated AuNCs and surface access for peroxidase-like activity in complex physiological environments. We subsequently developed a biosensor by encapsulating these optimised AuNCs in bacterial toxin-responsive liposomes, which was extensively studied by various single-particle techniques. Upon exposure to *S. aureus* toxins, the liposomes rupture, releasing AuNCs that generate a colourimetric signal after kidney-mimetic filtration. The biosensor was further validated *in vitro* and *in vivo* using a hyaluronic acid (HA) hydrogel implant infection model. Urine collected from mice with bacteria-infected HA hydrogel implants turned blue upon substrate addition, confirming the suitability of the sensor for non-invasive detection of implant-associated infections. This platform has significant potential as a versatile, cost-effective diagnostic tool.”~~

~~“*Staphylococcus aureus*, including methicillin-resistant *S. aureus* (MRSA), are responsible for a raft of minor and serious infections in humans and is the leading cause of surgical implant-associated infection¹.”~~

~~“SARS-CoV-2 and *S. aureus* co-infection has been exacerbating morbidity and mortality, prompting concerns about the need for early detection to improve treatment outcomes; particularly in cases with a heightened risk of Surgical site infections (SSI) are a major complication associated with procedures involving indwelling medical devices^{2,3}.”~~

~~“The treatment of implant infections is further complicated by drug-resistant pathogens such as methicillin-resistant *S. aureus* (MRSA), which is the second leading cause of mortality associated with antibiotic resistance worldwide⁴.”~~

“This approach facilitates the development of non-invasive, real-time monitoring systems for early diagnosis and personalised treatment strategies. Improvement in the catalytic activity of the underlying AuNCs could enhance sensitivity of the diagnostic to allow more accurate and earlier detection of pathological conditions. However, AuNC size must remain below the glomerular filtration cut-off (ca. 5.5 nm)³⁸ to ensure rapid clearance of catalytically optimised AuNCs through the kidneys. Mechanistic understanding of stabilising ligand distribution and behaviour around the AuNCs is another key aspect required to move forward the development of diagnostic tests using renal-clearable, catalytically active AuNCs.”

“This improvement in AuNC catalytic performance can enhance the sensitivity of the sensing platform, allowing for detection even with minimal liberation of AuNCs, which is particularly relevant when aiming to detect disease early. Compared to our previous work³², we improved the catalytic activity of AuNCs, changed the sensor design away from a covalent peptide-AuNC conjugate to a non-covalent liposomal system, expanded application from cancer to bacterial infection, and provided extensive mechanistic understanding by leveraging various single-particle techniques and computational analysis.”

3. The reviewer is sure that two technical repeats, such as the case of Fig. 5B, can show the trend of group differences, but is not sure that one or two repeats are enough to reach a statistical conclusion.

Thank you for reminding us of the importance of replicates. We have now repeated experiments that had less than three independent repeats (including the requested Figure 5b). All the conclusions of the manuscript remain the same as all the experiments were repeatable and delivered similar results.

Figure 5b

Figure 5. *In vitro* incubation of bacteria with AuNC-loaded liposomes and characterisation of bacterial growth. (a) Schematic of *in vitro* incubation of AuNC-loaded liposomes with bacteria and subsequent separation of released AuNCs from intact liposomes. (b) Bacterial growth curve of *S. aureus* SH1000 by CFU counting method (mean values \pm standard error, N = 3 independent repeats). (c-e) TMB oxidation assay on the filtrates after incubation of AuNC-loaded liposome with TSB +/- *S. aureus* from 0 to 24 hours. (c) Photographs of final TMB assay results. (d, e) Absorbance (at 652 nm at 140 second endpoint) and onset velocity for assay in (c): TSB (black bars) and TSB + bacteria (grey bars) (mean values \pm standard error, N = 3, n=1, additional repeat with detailed reaction curve in Supplementary Figure 19a-e). (f, g) *In vitro* cytocompatibility MTS cell viability assay after incubation of AuNCs (f) and AuNC-loaded liposomes (g) with RAW 264.7 cells (mean values \pm standard error, N = 3 individual experiments, n = 3 measurements).

4. Stability (in vitro and/or in vivo) of this liposomal AuNC system should be comprehensively tested.

Thank you for your insightful comment on this aspect and we completely agree. Reviewer #1, comment 2 has raised a similar point. Please see our answer, added data and discussion in our response to reviewer #1 comment. In summary, we have now provided a comprehensive study on stability for the liposomal AuNC system, including when embedded in the HA gel, using our *in vitro* and *in vivo* setups. All the experiments were now performed up to two days instead of the few hours presented before and it confirmed excellent stability of the sensor system, both *in vitro* and *in vivo*, in the absence of pathogenic bacteria.

5. “AuNC-liposomes mixed with HA hydrogel were either bacteria-free or intentionally contaminated with *S. aureus*.” Is the use of hydrogel a good simulation of patients with implant infection in real clinical settings? In addition, there is no validation of the (liner or other) responsiveness of the sensing system with the severity of *S. aureus* local infection. When considering *in vivo* stability and bacterial specificity of this sensing system (or the lack of them), the feasibility seems to be in heavy doubt.

We thank the reviewer for raising relevant points regarding the chosen model, sensitivity with respect to bacteria numbers, stability and specificity towards pathogenic bacteria. First, the HA hydrogel we have chosen is a commercial, clinically approved product that is used as dermal filler (Belotero intense)². Since *S. aureus* is one of the main causes of nosocomial implant-associated infection⁴, which is also a concern for dermal fillers, such as HA hydrogels⁵, we have selected our scenario. Bacterial infection of implants occurs due to unintended introduction of bacteria during the insertion process. Hence, we believe our scenario of injecting bacteria-contaminated HA hydrogels is clinically relevant. Second, regarding correlation of signal generation with bacterial numbers (severity of infection), our data regarding bacterial growth (CFU/mL over time 1:100, Figure 5b) can be compared to the TMB signal observed in the tested filtrates after incubation for different timeframes (Figure 5c-e; Figure 6 b,c; Supplementary Figures 19a-e and 22). We can already detect TMB signal after 2 h, indicating that ca. 10⁸ CFU/mL bacteria have produced enough toxin within 2 h to start causing liposomal rupture and release of AuNCs. The signal largely follows the bacteria growth curves and saturates after about 7 – 8 h. We associate this plateauing with the inability of AuNCs to readily pass through the Amicon filter due to lots of bacterial toxin/bacteria/liposomal debris blocking the membrane at later timepoints. This is supported by the finding that *in situ* FCS measurements, which does not require separation of liberated and encapsulated AuNCs, revealed that 99 % of AuNCs were released after 2 h of toxin incubation (Figure 4e). Third, regarding stability, we have now confirmed stability of up to two days, please see our response to your previous comment 4 (reviewer #1, comment 2). Fourth, to demonstrate bacterial specificity, we performed an additional experiment, incubating our liposomal sensors with supernatant from a non-pathogenic bacterial culture

(*Lactococcus lactis*). TMB assay was conducted in the filtrates after incubation and Amicon filter separation (Supplementary Figure 23). The results showed that only pathogenic *S. aureus* and not *L. lactis* triggered release of AuNCs from the liposomal sensor, confirming bacterial specificity.

“The commercial product we used (Belotero intense) is widely used and has a reported longevity of up to 12 months after injection⁷⁷. Infection of hydrogel dermal fillers through implantation of inadvertently contaminated implants is also a concern, similar to other implants⁷⁸.”

“When considering the bacteria numbers, we can estimate that about 10^8 CFU/mL produced enough toxin within 1 – 2 h to cause sufficient sensor rupture and AuNC release to be available for TMB signal generation. To evaluate the specificity of the liposomal sensor, we incubated sensors with supernatants from cultures of *S. aureus* and the non-pathogenic, non-toxin producing bacterium *Lactococcus lactis*. The TMB assay results indicated that the liposomal sensor specifically responded to the *S. aureus* toxin, as evidenced by the blue colour change observed in the filtrate containing released AuNCs. In contrast, no blue colour was visible in the filtrate from the incubation with *L. lactis*, demonstrating that the liposomal sensor selectively responded to *S. aureus* (Supplementary Figure 23 c-g). This is in agreement with a previous study that revealed specificity of liposomal rupture in presence of pathogenic *S. aureus* vs non-pathogenic *L. lactis*²².”

Supplementary Figure 23. Bacteria growth in presence of HA hydrogels and responses of the liposomal sensor to *S. aureus* and *L. lactis*. (a) Schematic of incubation of *S. aureus* SH1000 strain with and without HA hydrogel. (b, c) CFU measurements ($n = 1$ measurement) and absorbance at 600 nm measurements ($n = 3$ measurement) of the samples from SH1000 incubation with and without HA hydrogel demonstrating the gel had no effect on bacteria growth. (d) Photographs of TMB results for (g, h). (e, f) The calibration curve of the liposomal sensor by TMB assay, calculating the onset velocity (e) and measuring the endpoint absorbance (f). (g, h) TMB colourimetric readout results on the filtrates after Amicon filtration post 3-hour incubation of the liposomal sensor with supernatants from *S. aureus* and *L. lactis*, thrombin and the media controls (TSB and M 17 media), showing the absorbance (g), onset velocity (h).

6. Would the urine volume, urine color, or collection timing, affect the results? Would other diseases disturb the liposome and release the GSH-AuNC?

Thank you for your valuable comment. Indeed, urine volume and collection timing can influence the results. Depending on the amount of liquid consumed over the assay time, the urine volume will be different, hence the concentration of liberated AuNCs would differ. The same applies to collection timing. Depending on the local bacteria growth rate, which will depend on the initial level of contamination, the peak AuNC release might be occurring at different timepoints. However, we envision our assay to provide a binary yes/no answer rather than a quantitative readout. Hence, we need to ensure that sufficient sensor is present to allow detection of minimal AuNC amounts independent of urine volume changes and differences in timing. The colour of the urine does not have a significant impact, as the TMB substrate concentration is used in excess, we analyse signal increase over time (any background urine signal would remain constant over time) and yellow urine has low absorption at the TMB wavelength of 652 nm. Regarding other diseases, hence specificity of the sensor, we cannot currently exclude that any other disease cannot liberate the AuNCs. However, to provide some evidence of specificity we have now included a non-pathogenic bacteria control, which did not liberate AuNCs (please see our response to your comment 5). Additionally, we have picked thrombosis as an unrelated disease. Incubation of the liposomal sensor with thrombin did not cause any AuNC release (Supplementary Figure 23), providing further evidence of specificity towards pathogenic bacteria. If any disease condition would be found in the future to inadvertently cause AuNC release, we envision integration of a dual-response system could resolve this issue. The PEG corona on the liposomes could be increased beyond the current 1mol% to initially block toxin/other enzyme access. The PEG could be linked through enzyme-sensitive peptides which would create a system that first requires PEG removal through enzymatic cleavage (e.g. a host enzyme upregulated in infections or a specific bacterial enzyme) and only then allow toxin-mediated rupture of the liposome

(dual response). A similar strategy with cleavable PEG has been effective in improving specificity of drug delivery in the cancer field⁶. We have added the thrombin data and corresponding text to the manuscript and inserted discussion points regarding future improvement of specificity and clarified the importance of urine volume/timing of sampling.

“To evaluate the specificity of the liposomal sensor, we incubated sensors with supernatants from cultures of *S. aureus* and the non-pathogenic, non-toxin producing bacterium *Lactococcus lactis*. The TMB assay results indicated that the liposomal sensor specifically responded to the *S. aureus* toxin, as evidenced by the blue colour change observed in the filtrate containing released AuNCs. In contrast, no blue colour was visible in the filtrate from the incubation with *L. lactis*, demonstrating that the liposomal sensor selectively responded to *S. aureus* (Supplementary Figure 23 d-h). This is in agreement with a previous study that revealed specificity of liposomal rupture in presence of pathogenic *S. aureus* vs non-pathogenic *L. lactis*²². We have further selected thrombosis as an unrelated disease to investigate diagnostic specificity. After incubation of the liposomal sensor with thrombin, no colour change was detected when performing the TMB assay in the filtrate (Supplementary Figure 23), confirming the inability of the sensor to respond to this enzyme. If future research reveals the need to improve sensor specificity further, we envision that a dual-response system with an enzyme-cleavable PEG shell could be explored. This strategy has previously been successful in improving specificity in the anti-cancer drug delivery field⁷⁹.”

~~“In the current setting, only the highest signals were clearly observable as a blue colour by the naked eye. Further optimisation of AuNC catalytic activity, loading amount of AuNCs per liposomes, and liposome composition are future means to further optimise the detection sensitivity. Future studies on the effect of variations in urine volume (dependent on liquid uptake) and time window of urine collection on signal intensity is required to define the necessary minimal sensor concentration needed at the implant site to achieve the envisioned binary yes/no readout of bacterial infection status.”~~

7. In hypothetical clinical setting, how would this system being used? It is interesting to read that the authors proposed “a covalent coupling strategy to anchor the sensor to the implant to extend the duration of liposome retention within the hydrogel”, “but (the authors) envisage this as future work”.

The current system could be used clinically in the same way as demonstrated in our mouse model. The sensor could simply be mixed into the commercial hydrogel, before injection. This non-covalent system would then allow detection of acute implant infection. The patient could take a urine sample a few hours after the procedure, even when at home, and add it into a container with pre-mixed TMB substrate solution. If the

urine turns blue, the implant was infected and the patient could go to the GP for an antibiotic prescription. To allow detection of implant infection at later timepoints, and also for dense implants without available pockets for the liposomes, covalent coupling of the sensor to the implant would be required to ensure long-term localisation. Please see our response to reviewer #1 comment 2 where we have discussed future development of such a covalent system. We have also added a discussion point regarding the described envisioned clinical scenario using our current system.

“We envision the application of the current system for detecting acute implant infection. The clinical scenario could follow the same procedure as demonstrated in our animal experiments, involving mixing the sensor with the hydrogel before injection and collecting a urine sample from the patient a few hours after the procedure. If the urine turns blue upon addition of the TMB substrate solution, an antibiotic could be prescribed. Alternatively, the liposomal sensor could be injected locally in any area of suspected bacterial infection and urine could be collected at a defined later time point for sensing via the catalytic reaction. Further research and refinement of this biosensor platform could lead to its practical clinical application with particular relevance for at-home or resource limited settings due to its simple and sensitive colorimetric readout.”

8. The reviewer agrees that the authors modified this sensing system extensively with regards to its catalytic activity, mode of action, and underlying mechanism. But it seems that the results focus more on mechanical exploration other than echoing the introduction and the clinical question.

We thank the reviewer for acknowledging our extensive efforts to mechanistically understand our sensing system, which we believe is fundamental for the further development towards clinical application of this system. We agree that there was some disconnect between the introduction and the main text, which we have now improved by deleting some of the clinical information and focusing more on the mechanistic aspects. Please see the changes as highlighted in our response to your comment 2.

References

1. Loynachan, C. N. *et al.* Renal clearable catalytic gold nanoclusters for in vivo disease monitoring. *Nat Nanotechnol* **14**, 883–890 (2019).
2. Prasetyo, A. D., Prager, W., Rubin, M. G., Moretti, E. A. & Nikolis, A. Hyaluronic acid fillers with cohesive polydensified matrix for soft-tissue augmentation and rejuvenation: A literature review. *Clin Cosmet Investig Dermatol* **9**, 257–280 (2016).
3. Abarghohi, S., Fakhri, N., Borghei, Y. S., Hosseini, M. & Ganjali, M. R. A colorimetric paper sensor for citrate as biomarker for early stage detection of

- prostate cancer based on peroxidase-like activity of cysteine-capped gold nanoclusters. *Spectrochim Acta A Mol Biomol Spectrosc* **210**, 251–259 (2019).
4. Pietrocola, G. *et al.* Colonization and Infection of Indwelling Medical Devices by *Staphylococcus aureus* with an Emphasis on Orthopedic Implants. *Int J Mol Sci* **23**, (2022).
 5. Wang, Y. *et al.* In vitro and in vivo methods to study bacterial colonization of hydrogel dermal fillers. *J Biomed Mater Res B Appl Biomater* **110**, 1932–1941 (2022).
 6. Fang, Y. *et al.* Cleavable PEGylation: a strategy for overcoming the “PEG dilemma” in efficient drug delivery. *Drug Deliv* **24**, 22–32 (2017).